# GENERATIVE ADVERSARIAL TRAINING FOR NEURAL COMBINATORIAL OPTIMIZATION MODELS

## ABSTRACT

Recent studies show that deep neural networks can be trained to learn good heuristics for various Combinatorial Optimization Problems (COPs). However, it remains a great challenge for the trained deep optimization models to generalize to distributions different from the training one. To address this issue, we propose a general framework, Generative Adversarial Neural Combinatorial Optimization (GANCO) which is equipped with another deep model to generate training instances for the optimization model, so as to improve its generalization ability. The two models are trained alternatively in an adversarial way, where the generation model is trained by reinforcement learning to find instance distributions hard for the optimization model. We apply the GANCO framework to two recent deep combinatorial optimization models, i.e., Attention Model (AM) and Policy Optimization with Multiple Optima (POMO). Extensive experiments on various problems such as Traveling Salesman Problem, Capacitated Vehicle Routing Problem, and 0-1 Knapsack Problem show that GANCO can significantly improve the generalization ability of optimization models on various instance distributions, with little sacrifice of performance on the original training distribution.

## 1 INTRODUCTION

Combinatorial Optimization Problems (COPs) are a family of problems with the goal of finding the best one(s) from a finite set of solutions. Due to the considerably large solution space, many important COPs are hard to solve, such as the vehicle routing problems (Toth & Vigo, 2002). Exact algorithms based on branch-and-bound (Lawler & Wood, 1966) or its variants can provide elegant theoretical guarantees but the worst-case computational complexity is exponential, hence impractical for problems of medium and large sizes. In contrast, heuristic methods can usually attain good solutions in reasonable running time, which are often preferred in practical applications.

Traditional heuristics are designed based on expert knowledge for specific problems, which usually requires a large amount of time and efforts to develop. These manually designed heuristics could suffer from relatively poor performance. Moreover, for less studied problems, sufficient expert knowledge may even be unavailable. Recent studies suggest that deep learning could greatly facilitate in automating the design of heuristics, and alleviating the heavy reliance on expert knowledge. With the prior that the instances may follow certain distribution (e.g., locations in a routing problem may uniformly scatter in an area), deep models can be trained to learn heuristics in an end-to-end way (Dai et al., 2017; Kool et al., 2019; Chen & Tian, 2019). It has been shown that these models perform well with relatively short running time on COP instances following the training distribution.

However, after the trained model is deployed, it could encounter many instances following unknown distributions different from the training one, especially for real-life applications. As shown in many existing works and our experiments, when applied to infer the instances following a different distribution, the generalization performance of deep models gets much inferior, which severely hinders the practical use of the learned heuristics. Such mismatch between the training and testing distributions is always an important issue for most learning based methods. Especially, for neural combinatorial optimization (NCO) models, deep learning models are mostly trained on instances sampled from specific distributions and the solution quality intricately depends on the instance distributions. The generalization to instances with different distributions has been widely acknowledged for the importance (e.g., in Mazyavkina et al. (2021)) and remains a challenge.

To tackle this issue, we propose the Generative Adversarial Neural Combinatorial Optimization (GANCO) framework which is model agnostic and generally applicable to various neural combinatorial optimization models for solving different COPs. Instead of training an *optimization* model only on instances following the predefined distribution, another deep neural network is deployed as a *generation* model to produce training instances following the distributions on which the optimization model performs poorly. The generation model and optimization model are trained alternatively in an adversarial way. Specifically, the generation model is trained by reinforcement learning to maximize the performance gap of the current optimization model on the generated instances with respect to a traditional non-learning baseline algorithm. The optimization model is trained in the original way but using the training dataset augmented with the newly generated instances to improve the generalization performance, i.e., to reduce the performance gap. As we will show in the experiments, the non-learning baseline algorithms do not need to be very strong or fast.

To demonstrate the effectiveness of the proposed GANCO framework, we apply it to a representative neural combinatorial optimization model, the Attention Model (AM, Kool et al. (2019)) on various COPs including Traveling Salesman Problem (TSP), Capacitated Vehicle Routing Problem (CVRP), Orienteering Problem (OP), Prize Collecting TSP (PCTSP) and 0-1 Knapsack Problem (KP). In the extensive experiments, we show that the proposed GANCO framework improves the generalization performance of the original optimization model on various testing distributions with little sacrifice of performance on the original training distribution. Furthermore, we show that the proposed GANCO framework can be readily and effectively applied to other optimization models such as the Policy Optimization with Multiple Optima (POMO, Kwon et al. (2020)).

## 2 RELATED WORKS

Most deep learning models are trained to learn the construction heuristics where the model picks the actions sequentially to construct a solution (e.g., Bello et al. (2017); Nazari et al. (2018); Kool et al. (2019); Hottung et al. (2021); Kwon et al. (2020)), which perform well with fairly short running time. While some other models learn the improvement heuristics to locally refine existing solutions, which usually search over a large number of solutions and are relatively slow. Moreover, they often need to fit in frameworks specifically designed for different problems, such as 2-opt (d O Costa et al., 2020), large neighborhood search (Hottung & Tierney, 2020) and combinations of local operators (Lu et al., 2020). Most models are trained with reinforcement learning (RL) except for several ones with supervised learning, e.g., Vinyals et al. (2015) and Joshi et al. (2019). Though we use representative construction heuristic models with RL in the experiments, our proposed GANCO framework can also be applied to models of other types.

The most well-known models trained with adversary are the Generative Adversarial Networks (Goodfellow et al., 2014; Yang et al., 2018; Zhang et al., 2019), which serve to generate new data samples to imitate the training set. The generative network generates the sample candidates whereas the discriminative network classifies whether the sample is generated or genuine. In contrast, the goal of our proposed GANCO framework is to train the optimization model for better generalization ability where the generation model is used only to generate more informative training samples. The idea of using adversarial training to improve RL robustness has also been studied in other fields such as Atari games and robotics (Pinto et al., 2017; Khirodkar et al., 2018; Dennis et al., 2020). However, existing works in this direction are not directly applicable to our task due to its unique properties. The key is how to define the adversary, which is specific to the application domain. In the field of COPs, we define the adversary as a hard instance generator, which is evaluated by performance gaps of the optimization model with respect to a traditional non-learning baseline algorithm.

To the best of our knowledge, the most related work considering generating data samples for COPs is Liu et al. (2020) where the samples are generated by performing crossover and mutation to existing instances. The goal is to search better parameters configuration for given traditional solvers. Whereas our proposed framework is designed to generate instances with the guidance of deep reinforcement learning and improve the generalization performance for neural combinatorial optimization models. In our experiments, we also verify the necessity of our data generation method.

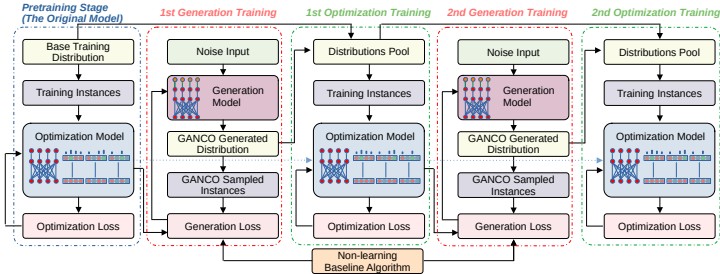

Figure 1: Generative Adversarial Neural Combinatorial Optimization (GANCO) framework. We use 2 adversarial iterations of generation training and optimization training for illustration purpose.

## 3 MODEL

For most learning based methods, the patterns are learned from the training data and usually applicable to similar testing samples. However, if the distributions of training and testing samples are considerably different, these learned patterns may not be general enough to guarantee desirable inference performance. As we will show in the experiments, this issue is particularly severe for COPs due to the intricate relationship between the solution and the instance distribution.

Instead of designing a network architecture to achieve better generalization performance for specific problems, we aim to propose a model agnostic framework which is generally applicable to different optimization models and training methods. The only requirement for the optimization models is that they are able to learn effective patterns from the training data and perform well on similar testing samples, which could be fulfilled by most of the recent learning based models.

To this end, we propose the Generative Adversarial Neural Combinatorial Optimization (GANCO) framework, as shown in Figure 1. Rather than training the optimization model solely on instances sampled from the predefined training distribution, we also deploy another deep learning model to generate instances on which the optimization model performs poorly. The generation model is trained by reinforcement learning to maximize the performance gap between the optimization model and a non-learning baseline algorithm on the generated instances. And the optimization model is trained with the generated samples to improve the performance on similar instances. In other words, the generation model and the optimization model are trained to maximize and minimize the performance gaps on the generated instances, respectively. In such an adversarial training way, the lower bound of the optimization model performance on the instance distributions generated by the generation model will be pushed up. We detail the generation model and the GANCO framework in the following subsections.

### 3.1 GENERATION MODEL

As suggested in existing works (Kool et al., 2019; Kwon et al., 2020), many important COPs can be viewed as a graph of $n$ nodes with different attributes, like the city coordinates for TSP and the item information for KP. Therefore, we formulate the data generation task as determining the attribute values for each node of an instance. However, the proposed framework could be easily adapted to other types of COPs, such as by determining the presence of each edge for problems depending on the graph connectivity (e.g., the Boolean Satisfiability Problem).

The generation model $\Omega$ takes the noise sample $h$ for each node as the input to output the distribution parameters for the node attributes. We format the noise input $h \in \mathbb{R}^{n \times 2}$ as 2-dimensional variables for each node $j = 1, .., n$ following the unit uniform distribution independently, i.e., $h \sim U(0, 1)$. Pertaining to the network architecture, we use the self-attention block proposed in the Attention Model (AM, Kool et al. (2019)), which consists of a multi-head self-attention layer (Vaswani et al., 2017), two node-wise feed-forward layers with Batch Normalization (Ioffe & Szegedy, 2015) and skip connection (He et al., 2016). The noise input is node-wise linearly projected to $H$-dimension and encoded by $k$ self-attention blocks. And another node-wise linear layer with a sigmoid activation projects the hidden vectors to the output distribution parameters $\Omega(h) \in \mathbb{R}^{n \times d}$ where $d$ is the number of attributes for each node. Then we draw the sample $\tilde{x}$ from the Gaussian distribution $\mathcal{N}(\mu, \sigma^2)$

with mean $\mu = \Omega(h)$ and fixed standard deviation $\sigma$. The network outputs are kept in the same range $[0, 1]$ to use the same variance for different attributes. The sample $\tilde{x}$ is scaled to the corresponding valid ranges (and discretized for discrete attributes) to attain the node attributes $x = A(\tilde{x})$.

The goal of the generation model is to generate instances on which the optimization model currently performs poorly. Since different instances have different optimal objective values, the attained objective values alone are not meaningful enough to evaluate the model performance. Although the optimality gap appears to be a good alternative metric in this case, it often requires expensive computation to attain the optimal solutions. On the other hand, while achieving approximate solutions, traditional non-learning algorithms tend to perform relatively stably on instances with different distributions, compared to the learning based methods. Therefore, the performance gaps of the optimization model with respect to a traditional non-learning baseline algorithm could be regarded as a favorable metric. Theoretically, the baseline algorithm performance can be arbitrarily inferior as long as it performs consistently (e.g., similar optimality gaps) on different instances. As we will show in our experiments, the baseline algorithms do not need to be very strong or fast as the training for generation models converges in a relatively small number of iterations.

The generation model is trained by the reinforcement learning (RL). Specifically, the *state* of RL environment is the noise input $h \in \mathbb{R}^{n \times 2}$. The *action* is to sample the attribute values for a node. An *episode* is to sample the attribute values $x = A(\tilde{x})$ with $\tilde{x} \sim \mathcal{N}(\Omega(h), \sigma^2)$ for all the nodes to form an instance. However, as the sampled attributes for one node will not affect the other nodes, the episode can be viewed as only containing one action with large action space to sample all the node attributes at the same time. The *reward* for an episode is the performance gap between the optimization model $\Phi$ and baseline algorithm $B$ on the sampled instance $x$. The loss function is formally defined as follow,

$$L(\Omega) = -E_{h \sim U(0,1), \tilde{x} \sim \mathcal{N}(\Omega(h), \sigma^2)} G(x, \Phi, B), \tag{1}$$

where $G(x, \Phi, B) = (O_\Phi(x) - O_B(x))/O_B(x)$ is the performance (objective) gap[1]; $O_\Phi(x)$ and $O_B(x)$ are the objective values of the solutions found by the optimization model $\Phi$ and the baseline algorithm $B$ for the generated instance $x$, respectively. Different from the generator in GAN (Goodfellow et al., 2014) which deterministically infers the instance given the noise input, we sample the noise input to infer the distribution and sample the instance from the output distribution. The generation model is trained by the REINFORCE algorithm (Sutton et al., 2000) with reward baseline as the mean of the gaps in a batch. The gradient to train the generation model is expressed as follow,

$$\nabla_\Omega L(\Omega) \approx -\frac{1}{N} \sum_{i=1}^{N} \left( G(x_i, \Phi, B) - \frac{1}{N} \sum_{j=1}^{N} G(x_j, \Phi, B) \right) \nabla_\Omega log \mathcal{N}\left( \tilde{x}_i; \Omega(h_i), \sigma^2 \right), \tag{2}$$

where $i$ and $j$ are the instance indices; $N$ is the batch size; $\tilde{x}_i \in R^{n \times d} \sim \mathcal{N}(\Omega(h_i), \sigma^2)$; and $\mathcal{N}(\tilde{x}_i; \Omega(h_i), \sigma^2)$ is the probability of $\tilde{x}_i$ sampled from the Gaussian distribution $\mathcal{N}(\Omega(h_i), \sigma^2)$.

## 3.2 GANCO FRAMEWORK

The formal flow of the GANCO framework is summarized in Algorithm 1. The GANCO framework is compatible with different optimization models and training algorithms. In the framework, the optimization model is always trained by the same algorithm from its original work.

**Pretraining Stage** First, we train the optimization model on the instances sampled from the original training distribution. To make sure that the performance improvement in the experiments is brought by the proposed GANCO framework rather than more epochs of training, we train the model until fully converged. However, convergence is not necessary in practice.

**Adversarial Training Stage** After pretraining, the generation model and optimization model are trained alternatively during the adversarial iterations. In each iteration, the generation model is first trained to generate the instance distributions on which the current optimization model exhibits poor performances. And the training distribution will be augmented with the newly generated distribution. Then the optimization model is trained with instances sampled from the augmented distribution to improve the performance on new instances while preserving the performance on old ones.

---

[1]Here we follow the convention to formulate the COPs as minimizing the objective functions. The maximization problem can be viewed as minimizing the negative objective.

**Generation Training** At the beginning of each adversarial iteration, the parameters of the generation model are re-initialized randomly. In doing so, it prevents the model from being stuck in the local optimum of previous adversarial iterations and also allows the model to learn diverse instance distributions. Specifically, the generation model is trained using the gradients in Eq. (2).

**Optimization Training** In each adversarial iteration, the optimization model is trained with the same algorithm from its original work but on different distributions. To achieve desirable performance on both the base training distributions and the adversarial generated distributions, we maintain a number of samples from the base distributions while adding the samples from the newly generated ones. In the experiments, we find that good performance could be easily and effectively realized by a simple data composition rule. Specifically, we always keep half of the samples from the base training distribution and then equally sample the other half from the generated distribution in each adversarial iteration. For optimization models trained by RL, we sample new instances from these distributions for each epoch following the composition rule above.

---

**Algorithm 1** Generative Adversarial Neural Combinatorial Optimization

---

**Input:** Optimization model $\Phi$, generation model $\Omega$, the non-learning baseline algorithm $B$, the base training distribution $D_o$, the number of adversarial iterations $N_a$, the number of training iterations $N_\Omega$ for $\Omega$ and the number of training epochs $N_\Phi$ for $\Phi$ in an adversarial iteration, the learning rate $\eta$ and batch size $N$ for $\Omega$.

**Output:** Generative adversarial trained optimization model $\Phi$

1: **while** $\Phi$ not converged **do**           ▷ Pretraining Stage
2:     Sample dataset $X$ from $D_o$, train $\Phi$ on dataset $X$
3: **end while**
4: Initialize the training distribution $D$ as $D_o$
5: **for** $k_a \leftarrow 1$ to $N_a$ **do**           ▷ Adversarial Training Stage
6:     Randomly initialize the parameters for $\Omega$
7:     **for** $k_\Omega \leftarrow 1$ to $N_\Omega$ **do**           ▷ $k_a$th Generation Training
8:        Sample the noise inputs $h \in \mathbb{R}^{N \times n \times 2}$, and infer the distribution parameters $\mu = \Omega(h)$
9:        **for** $i \leftarrow 1$ to $N$ **do**
10:          Sample the instances $x_i$ with $\tilde{x}_i \sim \mathcal{N}(\mu_i, \sigma^2)$, and use $\Phi$ and $B$ to solve $x_i$
11:        **end for**
12:        $\Omega \leftarrow \Omega - \eta \nabla_\Omega L(\Omega)$
13:     **end for**
14:     Augment $D$ with the distribution generated by $\Omega$
15:     **for** $k_\Phi \leftarrow 1$ to $N_\Phi$ **do**           ▷ $k_a$th Optimization Training
16:        Sample dataset $X$ from $D$ following dataset composition rule, and train $\Phi$ on dataset $X$
17:     **end for**
18: **end for**

---

## 4   Experiments and Analysis

We adopt the Attention Model (AM, Kool et al. (2019)) as the optimization model to verify the effectiveness of our proposed GANCO framework. AM is a widely used strong deep learning model for a series of routing problems and other COPs such as the Knapsack Problem. It is also used as the base model in many other recent works, e.g., Kwon et al. (2020); Xin et al. (2021); Hottung et al. (2021). As a heuristic model that learns to construct the solution sequentially, AM requires much less expert knowledge about the problems, especially compared with the region and rule design in Chen & Tian (2019) and the various perturbation operators in Lu et al. (2020). Typically, AM consists of an encoder and a decoder, which is trained by the REINFORCE algorithm with a deterministic rollout baseline. With three self-attention blocks, the encoder maps the nodes into embeddings. Based on the attention mechanism with glimpse, the decoder sequentially decides which node to visit at each step. We further improve AM in two ways, i.e., removing the Batch Normalization layers and training the model for more epochs until full convergence. Therefore, the performance of our base model is much better than the original ones reported in Kool et al. (2019). We discuss the reasons for the changes and the performance improvement with more details in Appendix H.

Table 1: Results of AM and GANCO-AM for TSP on instances with different distributions.

| Dist. | Method | $n = 20$ | | | $n = 50$ | | | $n = 100$ | | |
|---|---|---|---|---|---|---|---|---|---|---|
| | | Obj. | Gap | Time | Obj. | Gap | Time | Obj. | Gap | Time |
| *Base* | Concorde | 3.83 | 0.00% | 4s | 5.69 | 0.00% | 39s | 7.76 | 0.00% | 142s |
| | AM | 3.83 | 0.14% | 0.3s | **5.73** | **0.73%** | 0.8s | **7.93** | **2.16%** | 2.1s |
| | GANCO-AM | **3.83** | **0.14%** | 0.3s | 5.74 | 0.83% | 0.8s | 7.94 | 2.28% | 2.1s |
| *Cluster* | Concorde | 3.19 | 0.00% | 12s | 4.01 | 0.00% | 91s | 5.15 | 0.00% | 210s |
| | AM | 3.21 | 0.39% | 0.3s | 4.14 | 3.07% | 0.8s | 5.60 | 8.85% | 2.1s |
| | GANCO-AM | **3.20** | **0.27%** | 0.3s | **4.10** | **2.04%** | 0.8s | **5.46** | **6.06%** | 2.1s |
| *Uniform* | Concorde | 2.78 | 0.00% | 15s | 3.91 | 0.00% | 78s | 5.30 | 0.00% | 183s |
| | AM | 2.79 | 0.54% | 0.3s | 4.04 | 3.37% | 0.8s | 5.67 | 7.12% | 2.1s |
| | GANCO-AM | **2.78** | **0.19%** | 0.3s | **3.97** | **1.45%** | 0.8s | **5.51** | **3.94%** | 2.1s |
| *Diagonal* | Concorde | 2.43 | 0.00% | 21s | 2.72 | 0.00% | 152s | 3.24 | 0.00% | 196s |
| | AM | 2.49 | 2.29% | 0.3s | 3.35 | 23.21% | 0.8s | 5.22 | 61.47% | 2.1s |
| | GANCO-AM | **2.44** | **0.29%** | 0.3s | **2.78** | **2.37%** | 0.8s | **3.47** | **7.39%** | 2.1s |
| *Gaussian* | Concorde | 3.40 | 0.00% | 12s | 4.45 | 0.00% | 51s | 5.70 | 0.00% | 167s |
| | AM | 3.41 | 0.32% | 0.3s | 4.67 | 4.86% | 0.8s | 6.21 | 9.06% | 2.1s |
| | GANCO-AM | **3.41** | **0.27%** | 0.3s | **4.54** | **1.96%** | 0.8s | **6.10** | **7.02%** | 2.1s |
| *TSPLIB-S* | Concorde | 3.40 | 0.00% | 7s | 4.50 | 0.00% | 44s | 5.76 | 0.00% | 155s |
| | AM | 3.41 | 0.21% | 0.3s | 4.58 | 1.77% | 0.8s | 6.06 | 5.27% | 2.1s |
| | GANCO-AM | **3.41** | **0.17%** | 0.3s | **4.55** | **1.26%** | 0.8s | **5.99** | **4.12%** | 2.1s |

We use AM to solve TSP, CVRP, OP, PCTSP and KP in our experiments. The model is trained by the GANCO framework for 20 adversarial iterations, during each of which the generation model is trained for at most 2000 iterations (depending on problems) with batch size 100 and attribute standard deviation $\sigma$ 0.05, and the optimization model is trained for 20 epochs with 256000 instances in each. The hyper-parameters for model architecture and optimizer are the same as those in Kool et al. (2019). For fast convergence and stable training, we fix the parameters in AM encoder and only train the decoder after the pretraining stage. We leverage Concorde, Hybrid Genetic Search (HGS, Vidal (2020)), Compass (Kobeaga et al., 2018), Iterated Local Search (ILS) and Dynamic Programming (DP) for TSP, CVRP, OP, PCTSP and KP, respectively, as the non-learning baseline algorithms for training the generation model. As our GANCO framework is relatively robust to the performance of (non-learning) baseline algorithms, we set the number of non-improvement iterations before termination for HGS as 100 instead of the default 20000 to save training time. Also, we exploit a faster version of ILS and round the item weights to two decimal places for DP. For a fair comparison, we consider the node attribute ranges of the original training distribution as the valid ones and show the results for the five problems on various distributions of instances within those ranges.

Besides, we apply the GANCO framework to another prevailing optimization model, i.e., POMO (Kwon et al., 2020) to solve TSP and CVRP, and also present more analysis about our GANCO framework. All our codes will be made available soon.

## 4.1 Traveling Salesman Problem

For TSP, the attributes learned by the generation model are the two coordinates for each node. The base training instances are generated uniformly in the unit square, following Kool et al. (2019). We adopt instances following five different distributions as the testing sets to evaluate the generalization performance. Specifically, (1) *Clustered*: the nodes are distributed in clusters following Uchoa et al. (2017); (2) *Uniform*: the nodes are uniformly sampled in a rectangle with random aspect ratios; (3) *Diagonal*: the nodes are distributed near the diagonal of rectangle; (4) *Gaussian*: the nodes follow the bivariate Gaussian distribution with random correlation coefficients; (5) *TSPLIB-S*: the nodes are sampled without replacement from a random instance of the TSPLIB (Reinelt, 1991). We scale the instance coordinates to the valid range $[0, 1]$ with the aspect ratio fixed between horizontal and vertical axes so that the objective values of all solutions for an instance will only be scaled by the same constant. More details of data generation for each distribution are presented in Appendix A.

The results of Concorde, AM (with greedy decoding) and GANCO-AM are gathered in Table 1, where we report the average objective values and gaps, and the total time for solving the 10000 test-

Table 2: Results of AM and GANCO-AM for CVRP on instances with different distributions.

| Dist. | Method | $n = 20$ | | | $n = 50$ | | | $n = 100$ | | |
|---|---|---|---|---|---|---|---|---|---|---|
| | | Obj. | Gap | Time | Obj. | Gap | Time | Obj. | Gap | Time |
| *Base* | HGS | 6.11 | 0.00% | 0.5h | 10.34 | 0.00% | 1.4h | 15.57 | 0.00% | 3.2h |
| | AM | **6.28** | **2.74%** | 0.5s | **10.76** | **4.05%** | 1.2s | 16.38 | 5.22% | 2.8s |
| | GANCO-AM | 6.29 | 2.86% | 0.5s | 10.78 | 4.22% | 1.2s | **16.37** | **5.16%** | 2.8s |
| *Original* | HGS | 5.77 | 0.00% | 0.5h | 9.43 | 0.00% | 1.4h | 14.03 | 0.00% | 3.9h |
| | AM | **5.93** | **2.83%** | 0.5s | 9.90 | 5.03% | 1.2s | 15.05 | 7.25% | 2.7s |
| | GANCO-AM | 5.94 | 3.02% | 0.5s | **9.89** | **4.82%** | 1.2s | **14.90** | **6.18%** | 2.8s |
| *Small* | HGS | 3.55 | 0.00% | 0.2h | 5.11 | 0.00% | 1.0h | 7.21 | 0.00% | 3.1h |
| | AM | 4.08 | 14.86% | 0.5s | 5.82 | 13.87% | 1.2s | 8.79 | 21.80% | 3.4s |
| | GANCO-AM | **3.85** | **8.55%** | 0.4s | **5.63** | **10.05%** | 1.1s | **8.27** | **14.65%** | 2.6s |
| *Large* | HGS | 8.84 | 0.00% | 0.6h | 15.00 | 0.00% | 1.5h | 21.93 | 0.00% | 3.7h |
| | AM | 9.11 | 2.99% | 0.5s | 15.95 | 6.34% | 1.3s | 23.47 | 7.02% | 3.0s |
| | GANCO-AM | **9.04** | **2.17%** | 0.5s | **15.89** | **5.95%** | 1.2s | **23.29** | **6.20%** | 3.0s |
| *Identical* | HGS | 5.93 | 0.00% | 0.4h | 9.74 | 0.00% | 1.2h | 14.46 | 0.00% | 2.9h |
| | AM | 6.19 | 4.47% | 0.5s | 10.25 | 5.31% | 1.1s | 15.75 | 8.93% | 3.4s |
| | GANCO-AM | **6.12** | **3.34%** | 0.5s | **10.20** | **4.75%** | 1.1s | **15.47** | **7.03%** | 3.0s |
| *Quadrant* | HGS | 5.83 | 0.00% | 0.5h | 9.56 | 0.00% | 1.4h | 14.05 | 0.00% | 3.4h |
| | AM | 6.05 | 3.72% | 0.5s | 10.24 | 7.11% | 1.1s | 15.37 | 9.40% | 3.0s |
| | GANCO-AM | **6.04** | **3.59%** | 0.5s | **10.18** | **6.52%** | 1.2s | **15.14** | **7.76%** | 3.0s |
| *SL* | HGS | 4.40 | 0.00% | 0.4h | 6.69 | 0.00% | 1.3h | 9.52 | 0.00% | 3.4h |
| | AM | 4.59 | 4.18% | 0.4s | 7.17 | 7.21% | 1.1s | 10.67 | 12.06% | 3.3s |
| | GANCO-AM | **4.58** | **4.05%** | 0.4s | **7.13** | **6.59%** | 1.1s | **10.43** | **9.51%** | 2.8s |

ing instances sampled from each distribution, respectively. For the traditional algorithm Concorde, we run 20 instances in parallel on a 28-core CPU (also applies to HGS, Compass, ILS and DP in the subsequent subsections). The AM and GANCO-AM are trained and tested on a RTX-2080Ti GPU. Generally, their run time is much shorter than the traditional algorithm as indicated in the table(s), which is a core strength of deep models. Clearly, GANCO-AM improves the performance on various generalization testing sets (i.e., the last five distributions in Table 1) by large margins. While the optimality gaps of AM could be up to 61.47% for TSP100 on certain distribution, its performance after GANCO training tends to be much desirable and stable. As the GANCO framework does not change the optimization model architecture or the total number of parameters, the slight drop of performance on the base training distribution is fairly acceptable.

A more complete comparison with other methods is presented in Appendix A, where we show the performance improvement of GANCO-AM over AM while sampling more solution trajectories (e.g., GANCO-AM reduces the average gaps of AM on generalization distributions from 16.52% and 14.55% to 4.47% and 3.48% for TSP100 with 10 and 100 trajectories, respectively). Although we follow the convention in existing works (Kool et al., 2019) to train and test the model on fixed-size instances ($n = 20, 50, 100$ for routing problems), we also demonstrate that GANCO-AM significantly improves the results of AM on larger-size instances of the TSPLIB benchmark in Appendix B, where the average gap is reduced from 11.95% to 7.05% for instances with 100-300 nodes.

## 4.2 CAPACITATED VEHICLE ROUTING PROBLEM

Following Kool et al. (2019), the base training distribution with respect to the customer and depot coordinates is the same as the one for TSP. The demands are uniformly sampled from $\{1, ..., 9\}$, and the capacities are fixed as 30, 40, 50 for CVRP with 20, 50, 100 customers, respectively. Therefore, apart from the coordinate attribute, the generation model also learns the demand attribute given the fixed capacities. For the generalization testing sets, the customer nodes follow the six distributions used for TSP (including its base distribution). The depots are distributed uniformly, fixed at the center or a corner in the unit square, following CVRPLIB (Uchoa et al., 2017). For the customer demands, we adopt six distributions based on CVRPLIB. Particularly, they are sampled from $\{1, ..., 9\}$ (*Original*) or $\{1, 2\}$ (*Small*) or $\{8, 9\}$ (*Large*), random yet identical for all nodes within the same instance (*Identical*), small or large depending on the node quadrant (*Quadrant*), and most small with a few large ones (*SL*), respectively. More details for the testing sets are presented in Appendix C.

Table 3: Results of POMO and GANCO-POMO for CVRP100.

| Dist. | HGS | | Method | Single trajectory | | | 100 trajectories | | |
|-------|-----|-----|--------|------|-----|------|------|-----|------|
| | Obj. | Time | | Obj. | Gap | Time | Obj. | Gap | Time |
| *Base* | 15.57 | 3.2h | POMO | **16.09** | **3.32%** | 4.7s | **15.87** | **1.97%** | 51.3s |
| | | | GANCO-POMO | 16.12 | 3.56% | 4.9s | 15.89 | 2.09% | 51.6s |
| *Original* | 14.03 | 3.9h | POMO | 15.14 | 7.89% | 4.0s | 14.69 | 4.68% | 58.3s |
| | | | GANCO-POMO | **14.56** | **3.78%** | 3.8s | **14.35** | **2.26%** | 50.8s |
| *Small* | 7.21 | 3.1h | POMO | 10.15 | 40.74% | 4.1s | 8.84 | 22.47% | 61.1s |
| | | | GANCO-POMO | **7.78** | **7.82%** | 3.6s | **7.57** | **4.91%** | 48.0s |
| *Large* | 21.93 | 3.7h | POMO | 24.22 | 10.43% | 4.5s | 23.40 | 6.68% | 68.5s |
| | | | GANCO-POMO | **22.94** | **4.56%** | 3.9s | **22.58** | **2.95%** | 53.8s |
| *Identical* | 14.46 | 2.9h | POMO | 16.28 | 12.60% | 4.5s | 15.52 | 7.38% | 67.3s |
| | | | GANCO-POMO | **14.93** | **3.28%** | 4.1s | **14.75** | **2.01%** | 54.4s |
| *Quadrant* | 14.05 | 3.4h | POMO | 15.92 | 13.32% | 4.4s | 15.15 | 7.81% | 66.1s |
| | | | GANCO-POMO | **14.78** | **5.20%** | 3.8s | **14.51** | **3.24%** | 53.7s |
| *SL* | 9.52 | 3.4h | POMO | 10.79 | 13.28% | 4.2s | 10.25 | 7.67% | 57.9s |
| | | | GANCO-POMO | **10.05** | **5.52%** | 3.7s | **9.84** | **3.35%** | 49.1s |

As shown in Table 2, the proposed GANCO framework ameliorates the AM performance by large margins on almost all generalization testing sets with greedy decoding. However, for CVRP20, the performance of AM and GANCO-AM are close on some testing sets (such as *Original*, *Quadrant* and *SL*), which indicates that the original AM is fairly robust to certain distributions as the problem size is small. In Appendix C, we also compare with another popular traditional algorithm LKH3 (Helsgaun, 2017) and demonstrate the significant improvement of GANCO-AM over AM with the sampling (rather than greedy) version (e.g., the average gaps are reduced from 9.06% and 7.62% to 6.93% and 5.75% for CVRP100 with 10 and 100 trajectories, respectively).

### 4.3 Orienteering Problem, Prize Collecting TSP and 0-1 Knapsack Problem

We also apply the GANCO framework on AM to solve OP, PCTSP and KP, respectively. The base training distributions for OP and PCTSP are the same as those in Kool et al. (2019). The prizes setting for OP uses the hardest one reported in its original work, where the customer prizes increase with the *distance* to the depot. The base distribution for KP follows the setting in Kwon et al. (2020). More details for problem settings, generalization testing distributions and results are presented in Appendix D, E and F for OP, PCTSP and KP, respectively. GANCO significantly boosts the performance on the three problems of different sizes, with greedy decoding and trajectories sampling for AM, respectively (e.g., with greedy decoding, the average gaps for OP100, PCTSP100 and KP200 are reduced from 13.68%, 14.75% and 0.25% to 7.67%, 7.33% and 0.17%, respectively).

### 4.4 Policy Optimization With Multiple Optima

To demonstrate that the GANCO framework is model agnostic, we further apply it to the Policy Optimization with Multiple Optima (POMO, Kwon et al. (2020)) for solving TSP and CVRP. We only consider 100 nodes since the results of the improved AM in our experiments are superior or close to those of POMO on other sizes as reported in Kwon et al. (2020). In Table 3, we record the results of POMO and GANCO-POMO by inferring a single trajectory and 100 trajectories (with the multi-starting-node strategy) for solving CVRP100. Clearly, the improvement of GANCO-POMO over POMO is salient for both decoding methods. In Appendix G, we present more discussion about CVRP and detailed results for TSP100 (the average gaps are reduced from 7.79% to 2.08% with single trajectory, and from 3.64% to 0.90% with 100 trajectories for TSP100, respectively).

### 4.5 Analysis on the GANCO framework

In Figure 2, we plot the objective gaps of GANCO models on various testing sets along adversarial iterations for TSP or CVRP, where iteration 0 refers to the original AM or POMO. To better demonstrate the learning process, we adopt the gaps with respect to the best objective values among all the iterations. Though fluctuating on some testing sets especially for the early iterations, the

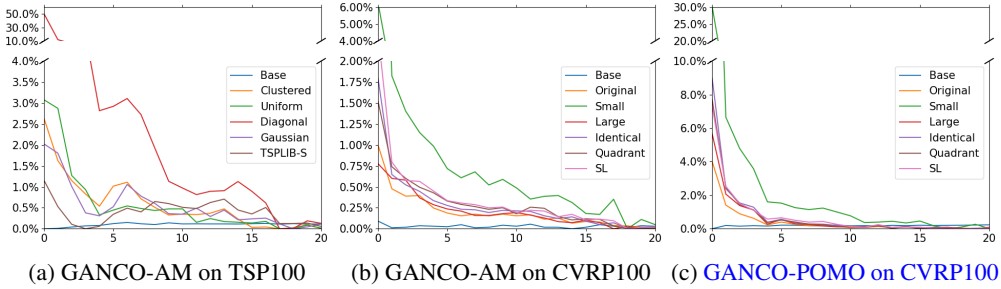

(a) GANCO-AM on TSP100    (b) GANCO-AM on CVRP100   (c) GANCO-POMO on CVRP100

Figure 2: Results of models trained with GANCO on testing sets along the adversarial iterations.

performance on those generalization testing sets becomes more desirable and stable quickly as the iteration increases.

Similarly, in Figure 3, we also plot the objective gaps of GANCO-AM for CVRP100 on 20 distributions, each of which is generated in the respective adversarial iteration. Clearly, the objective gaps of the worst-case instances found by the generation model are decreasing as the iteration increases, indicating that the lower bound of performance is pushed up with our GANCO. Furthermore, even before we train the model on a generated distribution (the diagonal polyline), the performance on it also usually tends to become better, which verified the effectiveness of the learned patterns. Besides, we also compare our data generation method with the genetic based one in Liu et al. (2020) to justify the deep RL in our GANCO. The details are presented in Appendix A.

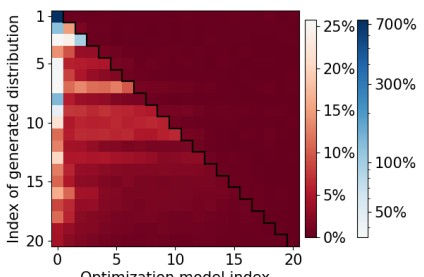

Figure 3: GANCO-AM results of CVRP100 on distributions generated by the generation model along the 20 adversarial iterations.

Though deep learning models still perform relatively inferior to strong traditional algorithms on well-studied COPs, GANCO framework effectively improves the generalization ability. With certain settings for the less studied problems, deep models performs favorably against the traditional ones (e.g., inferring 10 trajectories on PCTSP50, GANCO-AM improves the average gaps on testing sets from 2.32% to 1.31% with average run time 8s for 10000 instances, compared to the fast ILS 1.67% and 106s). Especially the adversarial training does not alter the model architecture and consumes fairly short time. The hardest training in our methods (GANCO-AM for CVRP100) costs less than 3 days compared to the 14 days for pretraining of AM. Furthermore, aiming at improving the generalization ability, GANCO could also be potentially applied to upcoming deep models, e.g., the one has stronger performance and could handle much larger sizes.

To further demonstrate the effectiveness of adversarial training in the proposed GANCO framework, we conduct more experiments in Appendix I. We demonstrate that GANCO achieves most of the benefits of training on in-distribution data and achieves better generalization performances compared to the model trained with multiple distributions. Furthermore, the generation model tailors distributions bespoke to the specific problem.

## 5   CONCLUSION AND FUTURE WORK

In this paper, we propose the GANCO framework to improve the generalization ability of Neural Combinatorial Optimization models. Trained by reinforcement learning, a generation model aims to find instances hard for an optimization model. With the two models trained alternatively in an adversarial way, the generalization ability of the optimization model is significantly boosted. We apply GANCO to two optimization models including AM and POMO on various COPs. Extensive experimental results show that our GANCO framework ameliorates the generalization performance by large margins with slight performance drops on the original training distribution. In future, we will investigate more optimization models and COPs, and study how to tackle much larger instances.

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

## A  AM EXPERIMENTS ON TSP

For the Traveling Salesman Problem (TSP), the base training instances are generated uniformly in the unit square (Kool et al., 2019). We adopt instances following five different distributions as the testing sets to evaluate the generalization performance.

1. *Clustered*: Following Uchoa et al. (2017), the nodes are distributed in clusters where the number of clusters $S$ is uniformly sampled from the integers $\{3, ..., 8\}$. With the unit square discretized to $1000 \times 1000$ grid, the cluster centers are uniformly and independently sampled. A node is located at location $p$ with probability proportional to $\sum_{s=i}^{S} exp(-w(p,s)/0.04)$, where $s$ is the cluster index and $w(p,s)$ is the distance between location $p$ and cluster center $s$. All nodes in an instance are located at distinct points in the grid.

2. *Uniform*: For each instance, the nodes are distributed uniformly in a rectangle. The aspect ratio of the rectangle is sampled from the uniform distribution $U(0,1)$ and the longer side is the horizontal or vertical axis with equal probabilities.

3. *Diagonal*: The nodes are distributed along the diagonal of a rectangle. For each instance, a rectangle is generated in the same way as in *Uniform* and a noise size $t$ is sampled from the uniform distribution $U(0.05, 0.2)$. One of the two diagonals is selected with equal probabilities. For each node in this instance, one point is sampled uniformly from the selected diagonal. And two noises independently sampled from the uniform distribution $U(-t, t)$ are imposed for the horizontal and vertical axes, respectively, to attain the node coordinates.

4. *Gaussian*: For each instance, a correlation coefficient $\rho$ is sampled from the uniform distribution $U(0,1)$. The nodes follow the bivariate Gaussian distribution with the mean $(0.5, 0.5)$, the standard deviation 1 for both axes, and the correlation coefficient $\rho$. However, we will scale the instance coordinates to the valid coordinate range $[0, 1]$. Therefore, the mean and standard deviation will not affect the scaled instance.

5. *TSPLIB-S*: For each instance, a reference instance is sampled from the TSPLIB benchmark (Reinelt, 1991). The nodes are sampled from this reference instance without replacement to form a new instance. We sample the instances in this way to have fixed number of nodes for testing set. However, we further present the results on the TSPLIB benchmark with various sizes in Appendix B.

We do not include the Random-Clustered node distribution in Uchoa et al. (2017), where half of the nodes are distributed uniformly in the unit square and the other half are distributed in 3-8 clusters. It is similar to the base unit uniform distribution for problems with 100 nodes or less.

In Table 4, we show the results of Concorde, LKH, original AM and GANCO-AM on the testing sets with different instance distributions, which include the base training distribution and 5 generalization testing distributions. Concorde is a highly optimized exact solver for TSP, which finds the optimal solutions in a fairly short time. LKH (Helsgaun, 2000) uses the same hyper-parameters (10000 trails and 10 runs) as those in Kool et al. (2019). For all the traditional algorithms (Concorde and LKH for TSP, HGS and LKH3 for CVRP, Compass for OP, ILS for PCTSP, DP for KP), we run 20 instances in parallel on a 28-core CPU. And the run time of AM and GANCO-AM is measured on a single RTX2080Ti GPU. In addition to the greedy decoding version of AM and GANCO-AM which infer one solution trajectory, we further include another two versions, i.e., inferring 10 trajectories (10t) and 100 trajectories (100t). In the greedy version, the node to visit at each step is selected as the one with the highest probability. More solution trajectories could be achieved by sampling the node to visit at each step for multiple times according to the inferred probability. With one solution trajectory greedily inferred, we further sample another 9 and 99 solution trajectories which lead to versions of 10 trajectories (10t) and 100 trajectories (100t), respectively. Clearly, for most settings, GANCO-AM improves the performance of the original AM by large margins. For small problem sizes ($n = 20$), some distributions (*Gaussian* and *TSPLIB-S*) are not significantly different from the base one, where the performance of AM and GANCO-AM is similar. For the base training distribution, the performance of GANCO-AM is fairly close to the original AM. As the GANCO framework does not change the model architecture or the total number of parameters, the small performance drop on the original training distribution is acceptable considering the improvement of generalization ability on other testing distributions.

To demonstrate the necessity of deep generation model trained by reinforcement learning in GANCO, we adopt the genetic based (GB) generation method in Liu et al. (2020) to our framework, namely GB-AM. Instead of managing instance distributions in GANCO, GB applies crossover and mutation operators to given instances for generating new ones. After the pretraining stage, GB first

Table 4: Results of AM and GANCO-AM for TSP on instances with different distributions.

| Dist. | Method | n = 20 | | | n = 50 | | | n = 100 | | |
|---|---|---|---|---|---|---|---|---|---|---|
| | | Obj. | Gap | Time | Obj. | Gap | Time | Obj. | Gap | Time |
| Base | Concorde | 3.83 | 0.00% | 4s | 5.69 | 0.00% | 39s | 7.76 | 0.00% | 142s |
| | LKH | 3.83 | 0.00% | 32s | 5.69 | 0.00% | 296s | 7.76 | 0.00% | 1241s |
| | AM greedy | 3.83 | 0.14% | 0.3s | 5.73 | 0.73% | 0.8s | 7.93 | 2.16% | 2.1s |
| | GANCO-AM greedy | 3.83 | 0.14% | 0.3s | 5.74 | 0.83% | 0.8s | 7.94 | 2.28% | 2.1s |
| | GB-AM greedy | **3.83** | **0.12%** | 0.3s | **5.73** | **0.71%** | 0.8s | **7.91** | **1.97%** | 2.1s |
| | AM 10t | 3.83 | 0.07% | 2s | 5.71 | 0.43% | 7s | 7.88 | 1.50% | 20s |
| | GANCO-AM 10t | 3.83 | 0.08% | 2s | 5.72 | 0.49% | 7s | 7.88 | 1.57% | 20s |
| | GB-AM 10t | **3.83** | **0.07%** | 2s | **5.71** | **0.40%** | 7s | **7.86** | **1.34%** | 20s |
| | AM 100t | 3.83 | 0.04% | 25s | 5.70 | 0.26% | 74s | 7.84 | 1.08% | 206s |
| | GANCO-AM 100t | 3.83 | 0.05% | 25s | 5.71 | 0.32% | 74s | 7.85 | 1.13% | 206s |
| | GB-AM 100t | **3.83** | **0.04%** | 25s | **5.70** | **0.25%** | 74s | **7.83** | **0.96%** | 206s |
| Cluster | Concorde | 3.19 | 0.00% | 12s | 4.01 | 0.00% | 91s | 5.15 | 0.00% | 210s |
| | LKH | 3.19 | 0.00% | 110s | 4.01 | 0.00% | 655s | 5.15 | 0.00% | 2754s |
| | AM greedy | 3.21 | 0.39% | 0.3s | 4.14 | 3.07% | 0.8s | 5.60 | 8.85% | 2.1s |
| | GANCO-AM greedy | **3.20** | **0.27%** | 0.3s | **4.10** | **2.04%** | 0.8s | **5.46** | **6.06%** | 2.1s |
| | GB-AM greedy | 3.21 | 0.40% | 0.3s | 4.14 | 3.22% | 0.8s | 5.64 | 9.53% | 2.1s |
| | AM 10t | 3.20 | 0.20% | 2s | 4.09 | 1.99% | 7s | 5.53 | 7.43% | 20s |
| | GANCO-AM 10t | **3.20** | **0.14%** | 2s | **4.06** | **1.25%** | 7s | **5.39** | **4.64%** | 20s |
| | GB-AM 10t | 3.20 | 0.22% | 2s | 4.10 | 2.11% | 7s | 5.57 | 8.24% | 20s |
| | AM 100t | 3.20 | 0.12% | 25s | 4.07 | 1.31% | 77s | 5.46 | 6.04% | 208s |
| | GANCO-AM 100t | **3.20** | **0.09%** | 25s | **4.05** | **0.81%** | 77s | **5.33** | **3.58%** | 208s |
| | GB-AM 100t | 3.20 | 0.12% | 25s | 4.07 | 1.42% | 77s | 5.50 | 6.85% | 208s |
| Uniform | Concorde | 2.78 | 0.00% | 15s | 3.91 | 0.00% | 78s | 5.30 | 0.00% | 183s |
| | LKH | 2.78 | 0.00% | 111s | 3.91 | 0.00% | 712s | 5.30 | 0.00% | 2051s |
| | AM greedy | 2.79 | 0.54% | 0.3s | 4.04 | 3.37% | 0.8s | 5.67 | 7.12% | 2.1s |
| | GANCO-AM greedy | **2.78** | **0.19%** | 0.3s | **3.97** | **1.45%** | 0.8s | **5.51** | **3.94%** | 2.1s |
| | GB-AM greedy | 2.79 | 0.53% | 0.3s | 4.12 | 5.25% | 0.8s | 5.67 | 7.06% | 2.1s |
| | AM 10t | 2.78 | 0.28% | 2s | 4.00 | 2.13% | 7s | 5.61 | 5.99% | 21s |
| | GANCO-AM 10t | **2.78** | **0.10%** | 2s | **3.95** | **0.83%** | 7s | **5.45** | **2.97%** | 21s |
| | GB-AM 10t | 2.78 | 0.26% | 2s | 4.05 | 3.54% | 7s | 5.61 | 5.96% | 21s |
| | AM 100t | 2.78 | 0.15% | 26s | 3.97 | 1.43% | 77s | 5.56 | 4.97% | 208s |
| | GANCO-AM 100t | **2.78** | **0.05%** | 26s | **3.93** | **0.51%** | 77s | **5.42** | **2.26%** | 208s |
| | GB-AM 100t | 2.78 | 0.14% | 26s | 4.01 | 2.40% | 77s | 5.56 | 4.96% | 208s |
| Diagonal | Concorde | 2.43 | 0.00% | 21s | 2.72 | 0.00% | 152s | 3.24 | 0.00% | 196s |
| | LKH | 2.43 | 0.00% | 472s | 2.72 | 0.00% | 1775s | 3.24 | 0.01% | 3371s |
| | AM greedy | 2.49 | 2.29% | 0.3s | 3.35 | 23.21% | 0.8s | 5.22 | 61.47% | 2.1s |
| | GANCO-AM greedy | **2.44** | **0.29%** | 0.3s | **2.78** | **2.37%** | 0.8s | **3.47** | **7.39%** | 2.1s |
| | GB-AM greedy | 2.48 | 1.90% | 0.3s | 3.66 | 34.72% | 0.8s | 5.78 | 78.76% | 2.1s |
| | AM 10t | 2.46 | 1.10% | 2s | 3.17 | 16.68% | 7s | 5.08 | 56.92% | 20s |
| | GANCO-AM 10t | **2.44** | **0.13%** | 2s | **2.76** | **1.47%** | 7s | **3.44** | **6.25%** | 20s |
| | GB-AM 10t | 2.46 | 0.96% | 2s | 3.26 | 19.89% | 7s | 5.61 | 73.51% | 20s |
| | AM 100t | 2.45 | 0.62% | 26s | 3.04 | 11.95% | 77s | 4.89 | 51.19% | 209s |
| | GANCO-AM 100t | **2.43** | **0.06%** | 26s | **2.74** | **0.88%** | 77s | **3.40** | **5.05%** | 209s |
| | GB-AM 100t | 2.45 | 0.53% | 26s | 3.06 | 12.65% | 77s | 5.34 | 65.02% | 209s |
| Gaussian | Concorde | 3.40 | 0.00% | 12s | 4.45 | 0.00% | 51s | 5.70 | 0.00% | 167s |
| | LKH | 3.40 | 0.00% | 75s | 4.45 | 0.00% | 421s | 5.70 | 0.00% | 1335s |
| | AM greedy | 3.41 | 0.32% | 0.3s | 4.67 | 4.86% | 0.8s | 6.21 | 9.06% | 2.1s |
| | GANCO-AM greedy | **3.41** | **0.27%** | 0.3s | **4.54** | **1.96%** | 0.8s | **6.10** | **7.02%** | 2.1s |
| | GB-AM greedy | 3.41 | 0.36% | 0.3s | 4.65 | 4.30% | 0.8s | 6.25 | 9.69% | 2.1s |
| | AM 10t | 3.41 | 0.16% | 2s | 4.61 | 3.41% | 7s | 6.15 | 7.92% | 20s |
| | GANCO-AM 10t | **3.41** | **0.15%** | 2s | **4.51** | **1.22%** | 7s | **6.01** | **5.54%** | 20s |
| | GB-AM 10t | 3.41 | 0.19% | 2s | 4.59 | 2.99% | 7s | 6.20 | 8.73% | 20s |
| | AM 100t | **3.40** | **0.08%** | 26s | 4.56 | 2.34% | 77s | 6.09 | 6.90% | 209s |
| | GANCO-AM 100t | 3.40 | 0.10% | 26s | **4.49** | **0.79%** | 77s | **5.95** | **4.33%** | 209s |
| | GB-AM 100t | 3.40 | 0.10% | 26s | 4.55 | 2.09% | 77s | 6.14 | 7.84% | 209s |
| TSPLIB-S | Concorde | 3.40 | 0.00% | 7s | 4.50 | 0.00% | 44s | 5.76 | 0.00% | 155s |
| | LKH | 3.40 | 0.00% | 40s | 4.50 | 0.01% | 296s | 5.76 | 0.01% | 1386s |
| | AM greedy | 3.41 | 0.21% | 0.3s | 4.58 | 1.77% | 0.8s | 6.06 | 5.27% | 2.1s |
| | GANCO-AM greedy | **3.41** | **0.17%** | 0.3s | **4.55** | **1.26%** | 0.8s | **5.99** | **4.12%** | 2.1s |
| | GB-AM greedy | 3.41 | 0.21% | 0.3s | 4.58 | 1.77% | 0.8s | 6.09 | 5.79% | 2.1s |
| | AM 10t | 3.40 | 0.11% | 2s | 4.55 | 1.23% | 7s | 6.01 | 4.33% | 21s |
| | GANCO-AM 10t | **3.40** | **0.10%** | 2s | **4.53** | **0.77%** | 7s | **5.93** | **2.94%** | 21s |
| | GB-AM 10t | 3.40 | 0.11% | 2s | 4.55 | 1.24% | 7s | 6.04 | 4.88% | 21s |
| | AM 100t | **3.40** | **0.06%** | 26s | 4.54 | 0.89% | 77s | 5.97 | 3.63% | 210s |
| | GANCO-AM 100t | 3.40 | 0.06% | 26s | **4.52** | **0.50%** | 77s | **5.88** | **2.18%** | 210s |
| | GB-AM 100t | 3.40 | 0.06% | 26s | 4.54 | 0.91% | 77s | 6.00 | 4.16% | 210s |

samples a dataset from the base training distribution. During the adversarial training stage, GB generates each new instance with the crossover and mutation operators. Two reference instances are randomly selected from the dataset. The $i$th node attributes (coordinates) are taken from the corresponding node of either reference instance with equal probability. And then each coordinate is

mutated with probability $1/\sqrt{n}$ to be replaced with a random number sampled from the unit uniform distribution ($n$ is the number of nodes). And same as GANCO, the objective gaps of the optimization model with respect to the baseline algorithm (Concorde) are used as the score for the instances. Given the objective gap scores, a binary tournament is used to remove an instance with lower score (since we plan to reserve the hard instances) and keep the dataset size unchanged. The genetic based generation is performed at the beginning of each adversarial iteration and the optimization model is trained on the new dataset. For a fair comparison, we use the same hyperparameters (number of adversarial iterations, number of running for the baseline algorithm, number of epochs for training the optimization model and number of instances in an epoch) for GB as those for GANCO.

As shown in Table 4, GB-AM does not improve the performance of AM on the generalization testing distributions. However, the performance on the base training distribution is slightly boosted. This result is expected because the genetic based generation method tends to search locally within the field of base training distribution to find the hard instances. The resulting distribution still approximately follows the base training distribution. It would be extremely hard for the genetic operators to attain instances following a significantly different distribution. If we discretize the node attributes, the generation task for finding instances with large objective gaps is essentially a Combinatorial Optimization Problem. Therefore, to tackle such a relatively hard task, we exploit a deep model trained with reinforcement learning, which is fairly effective as we have demonstrated in the experiments.

## B   AM EXPERIMENTS ON TSPLIB

In Table 5, we show the greedy decoding results of AM and GANCO-AM on TSPLIB instances with 100-1000 nodes. Both models are trained only with instances of 100 nodes. The instance coordinates are scaled to the valid range $[0, 1]$ with the fixed aspect ratio between the horizontal and vertical axes. Therefore, the objective values of the tours for an instance are scaled by the same constant. As the models are trained for 2-dimensional Euclidean TSP, we treat all the instances as in this distance space. Even though both models are trained with fixed-size graphs of 100 nodes, the performance on medium-size instances with 100-300 nodes are reasonably good. The average objective gaps for AM and GANCO-AM are 11.95% and 7.05% on instances with 100-300 nodes. Clearly, GANCO-AM effectively improves the generalization performance. However, both models start to deteriorate with lager sizes. The average gaps for AM and GANCO-AM are 27.44% and 23.37% on large instances with 300-1000 nodes. Nevertheless, the GANCO-AM still improves the generalization performance over AM.

## C   AM EXPERIMENTS ON CVRP

We detail the instance distributions for CVRP. The base training distribution is acquired from Kool et al. (2019), where the coordinates of customers and depot follow the unit uniform distribution, and the demands are uniformly sampled from the integers $\{1, ..., 9\}$. The capacity is fixed as 30, 40 and 50 for CVRP with 20, 50 and 100 nodes, respectively. Existing benchmarks such as CVRPLIB (Uchoa et al., 2017) have ranges of node attributes (demands and capacity) different from those of the base training distribution (the valid attribute ranges). Therefore, we use the following testing distributions based on the CVRPLIB benchmark. For all the generalization testing sets, the coordinates of customer nodes follow one of the 6 distributions for TSP (including the base training distribution and 5 generalization testing distributions) with equal probabilities. And the depot follows one of the 3 positioning with equal probabilities, 1) *Central*: at the center of unit square (0.5, 0.5), 2) *Eccentric*: at one of the four corners in the unit square, 3) *Random*: uniformly sampled from the unit square. The capacity is the same constant as the base training distribution. The demand distributions are adapted from the CVRPLIB, and presented as follows,

1. *Original*: The demands are uniformly sampled from $\{1, ..., 9\}$.

2. *Small*: The demands are sampled from the small integers $\{1, 2\}$.

3. *Large*: The demands are sampled from the large integers $\{8, 9\}$.

4. *Identical*: The demands for the nodes in the same instance are an identical integer uniformly sampled from $\{1, ..., 9\}$.

Table 5: Results of AM and GANCO-AM for TSP on TSPLIB instances.

| Instance | Opt. | AM Obj. | AM Gap | GANCO-AM Obj. | GANCO-AM Gap | Instance | Opt | AM Obj. | AM Gap | GANCO-AM Obj. | GANCO-AM Gap |
|---|---|---|---|---|---|---|---|---|---|---|---|
| KroA100 | 5.41 | 5.60 | 3.58% | **5.57** | **2.94%** | KroB100 | 5.63 | 5.88 | 4.45% | **5.83** | **3.68%** |
| KroC100 | 5.29 | 5.54 | 4.72% | **5.39** | **1.88%** | KroD100 | 5.46 | **5.58** | **2.09%** | 5.62 | 2.98% |
| KroE100 | 5.62 | 5.91 | 5.08% | **5.91** | **5.01%** | rd100 | 8.07 | 8.32 | 3.20% | **8.31** | **3.07%** |
| eil101 | 8.65 | **8.87** | **2.58%** | 8.94 | 3.31% | lin105 | 4.76 | **4.94** | **3.95%** | 5.12 | 7.64% |
| pr107 | 5.37 | 5.57 | 3.71% | **5.50** | **2.46%** | pr124 | 6.26 | 6.44 | 2.90% | **6.43** | **2.59%** |
| bier127 | 6.94 | 7.55 | 8.84% | **7.30** | **5.16%** | ch130 | 8.83 | 9.20 | 4.14% | **9.14** | **3.47%** |
| pr136 | 8.54 | 8.84 | 3.45% | **8.74** | **2.33%** | gr137* | 5.68 | 6.19 | 8.97% | **6.00** | **5.57%** |
| pr144 | 5.36 | **5.51** | **2.75%** | 5.59 | 4.34% | kroB150 | 6.66 | 6.95 | 4.36% | **6.89** | **3.46%** |
| kroA150 | 6.71 | 6.96 | 3.71% | **6.93** | **3.23%** | ch150 | 9.34 | 9.72 | 4.08% | **9.59** | **2.70%** |
| pr152 | 5.37 | 6.02 | 12.05% | **5.73** | **6.73%** | u159 | 8.09 | 8.72 | 7.80% | **8.30** | **2.63%** |
| rat195 | 8.08 | 9.12 | 12.96% | **8.99** | **11.29%** | d198 | 3.92 | 6.12 | 56.04% | **5.04** | **28.51%** |
| kroA200 | 7.45 | **7.91** | **6.09%** | 8.05 | 8.08% | kroB200 | 7.47 | **7.93** | **6.20%** | 7.99 | 6.98% |
| gr202* | 4.78 | 7.31 | 52.79% | **5.36** | **12.13%** | ts225 | 10.55 | **10.98** | **4.05%** | 11.21 | 6.23% |
| tsp225 | 8.22 | 9.09 | 10.62% | **9.06** | **10.28%** | pr226 | 5.54 | **5.76** | **3.94%** | 5.80 | 4.66% |
| gr229* | 4.63 | 9.04 | 95.43% | **5.81** | **25.52%** | gil262 | 12.05 | **12.58** | **4.42%** | 12.66 | 5.08% |
| pr264 | 6.38 | 7.50 | 17.59% | **7.19** | **12.61%** | a280 | 9.24 | 10.61 | 14.84% | **10.42** | **12.78%** |
| pr299 | 7.22 | **8.16** | **13.09%** | 8.18 | 13.36% | linhp318 | 10.17 | **11.09** | **9.09%** | 11.14 | 9.54% |
| lin318 | 10.17 | **11.09** | **9.09%** | 11.14 | 9.54% | rd400 | 15.34 | 17.41 | 13.48% | **17.04** | **11.05%** |
| fl417 | 6.29 | 8.28 | 31.66% | **7.05** | **12.12%** | gr431* | 5.45 | 9.39 | 72.38% | **7.45** | **36.83%** |
| pr439 | 8.99 | **10.72** | **19.27%** | 11.39 | 26.73% | pcb442 | 13.36 | 15.58 | 16.61% | **15.19** | **13.68%** |
| d493 | 9.35 | 12.51 | 33.75% | **11.48** | **22.80%** | att532 | 10.09 | **12.61** | **24.98%** | 12.62 | 25.06% |
| ali535* | 6.00 | 8.33 | 38.87% | **8.02** | **33.57%** | u574 | 12.02 | 14.71 | 22.34% | **14.67** | **21.98%** |
| rat575 | 13.62 | 16.75 | 22.97% | **16.66** | **22.36%** | p654 | 7.20 | **9.08** | **26.23%** | 9.71 | 34.97% |
| d657 | 12.21 | 15.30 | 25.27% | **14.96** | **22.50%** | gr666* | 8.74 | 11.65 | 33.32% | **11.65** | **33.26%** |
| u724 | 14.44 | 18.47 | 27.91% | **18.23** | **26.27%** | rat783 | 15.41 | 20.32 | 31.91% | **19.70** | **27.88%** |
| dsj1000 | 15.36 | 20.70 | 34.72% | **20.06** | **30.59%** | | | | | | |

\* The gr and ali535 instances are treated as in 2-D Euclidean space with their latitude and longitude as coordinates.

5. *Quadrant*: The demands depend on the quadrant of nodes. With the center of unit square $(0.5, 0.5)$ as the origin, the demands of nodes in even quadrants are sampled from small integers $\{1, 2\}$ while the demands of nodes in odd quadrants are sampled from large integers $\{8, 9\}$.

6. *SL*: Most nodes have small demands while other nodes have large ones. For one instance, a random percentage $p$ is sampled from the uniform distribution $U(70\%, 95\%)$. Thus, $p$ of the total nodes have demands sampled from the small integers $\{1, 2\}$ and the others have demands sampled from the large integers $\{8, 9\}$.

In Table 6, we show the results of HGS, a faster version of HGS, LKH3, the original AM and GANCO-AM on the various instance distributions. Hybrid Genetic Search (HGS, Vidal et al. (2012); Vidal (2020)) combines genetic algorithm with local search, which is considered as the state-of-the-art algorithm for CVRP. The faster version of HGS (HGS fast) uses the number of non-improvement iterations before termination as 100 instead of the default 20000 for HGS. HGS fast is the non-learning baseline algorithm which we use to train the generation model. LKH3 (Helsgaun, 2017) is an extension of LKH (Helsgaun, 2000) to solve various routing problems including CVRP. We use the same hyper-parameters (SPECIAL parameters, 10000 trials, 1 run) as those used in Kool et al. (2019). For distributions with small demands (*Small*) where the number of nodes in a route is large, LKH3 excels at the tour optimization within the route and outperforms HGS. However, LKH3 is unfriendly to the instances with many large-demand nodes. For these distributions (*Large*, *Identical*, *Quadrant*), LKH3 cannot find feasible solutions for some instances within 10000 trails.

We report the results of AM and GANCO-AM with greedy inferring, 10 trajectories (10t) and 100 trajectories (100t), respectively. As shown in Table 6, the GANCO-AM framework significantly improves the generalization performance over AM on most generalization testing settings. For small size ($n = 20$), GANCO-AM and AM exhibit similar performance on some distributions (*Quadrant* and *SL*), suggesting that AM is relatively robust to these distributions. And the improvement of GANCO-AM on all the generalization testing settings with relatively larger sizes ($n = 50, 100$) is significant.

Table 6: Results of AM and GANCO-AM for CVRP on instances with different distributions.

| Dist. | Method | n = 20 | | | n = 50 | | | n = 100 | | |
|---|---|---|---|---|---|---|---|---|---|---|
| | | Obj. | Gap | Time | Obj. | Gap | Time | Obj. | Gap | Time |
| Base | HGS | 6.11 | 0.00% | 1803s | 10.34 | 0.00% | 5043s | 15.57 | 0.00% | 11686s |
| | HGS fast | 6.12 | 0.03% | 14s | 10.35 | 0.10% | 52s | 15.69 | 0.80% | 153s |
| | LKH3 | 6.12 | 0.07% | 3336s | 10.35 | 0.10% | 13223s | 15.65 | 0.54% | 24393s |
| | AM greedy | **6.28** | **2.74%** | 0.5s | **10.76** | **4.05%** | 1.2s | 16.38 | 5.22% | 2.8s |
| | GANCO-AM greedy | 6.29 | 2.86% | 0.5s | 10.78 | 4.22% | 1.2s | **16.37** | **5.16%** | 2.8s |
| | AM 10t | **6.23** | **1.87%** | 3s | **10.64** | **2.84%** | 10s | 16.18 | 3.92% | 25s |
| | GANCO-AM 10t | 6.23 | 1.92% | 3s | 10.64 | 2.90% | 10s | **16.17** | **3.87%** | 25s |
| | AM 100t | **6.20** | **1.36%** | 40s | **10.57** | **2.18%** | 107s | 16.06 | 3.14% | 265s |
| | GANCO-AM 109t | 6.20 | 1.41% | 40s | 10.57 | 2.18% | 109s | **16.05** | **3.11%** | 268s |
| Original | HGS | 5.77 | 0.00% | 1775s | 9.43 | 0.00% | 4926s | 14.03 | 0.00% | 13921s |
| | HGS fast | 5.77 | 0.05% | 14s | 9.44 | 0.11% | 51s | 14.12 | 0.65% | 155s |
| | LKH3 | 5.77 | 0.07% | 3498s | 9.44 | 0.07% | 14042s | 14.08 | 0.33% | 29169s |
| | AM greedy | **5.93** | **2.83%** | 0.5s | 9.90 | 5.03% | 1.2s | 15.05 | 7.25% | 2.7s |
| | GANCO-AM greedy | 5.94 | 3.02% | 0.5s | **9.89** | **4.82%** | 1.2s | **14.90** | **6.18%** | 2.8s |
| | AM 10t | **5.87** | **1.81%** | 3s | 9.76 | 3.53% | 10s | 14.86 | 5.89% | 26s |
| | GANCO-AM 10t | 5.88 | 1.91% | 3s | **9.74** | **3.30%** | 10s | **14.71** | **4.84%** | 26s |
| | AM 100t | **5.84** | **1.25%** | 40s | 9.68 | 2.66% | 111s | 14.72 | 4.89% | 272s |
| | GANCO-AM 109t | 5.85 | 1.34% | 40s | **9.66** | **2.44%** | 111s | **14.59** | **3.96%** | 268s |
| Small | HGS | 3.55 | 0.10% | 823s | 5.11 | 0.19% | 3549s | 7.21 | 0.01% | 11073s |
| | HGS fast | 3.55 | 0.10% | 11s | 5.12 | 0.31% | 40s | 7.25 | 0.53% | 146s |
| | LKH3 | 3.55 | 0.00% | 153s | 5.10 | 0.00% | 1632s | 7.21 | 0.00% | 4366s |
| | AM greedy | 4.08 | 14.98% | 0.5s | 5.82 | 14.08% | 1.2s | 8.79 | 21.82% | 3.4s |
| | GANCO-AM greedy | **3.85** | **8.67%** | 0.4s | **5.63** | **10.26%** | 1.1s | **8.27** | **14.66%** | 2.6s |
| | AM 10t | 3.91 | 10.23% | 3s | 5.62 | 10.21% | 10s | 8.48 | 17.52% | 31s |
| | GANCO-AM 10t | **3.74** | **5.59%** | 3s | **5.46** | **6.94%** | 9s | **8.08** | **11.97%** | 26s |
| | AM 100t | 3.80 | 7.18% | 38s | 5.49 | 7.51% | 114s | 8.27 | 14.59% | 312s |
| | GANCO-AM 109t | **3.69** | **3.96%** | 36s | **5.36** | **4.94%** | 102s | **7.93** | **9.97%** | 261s |
| Large | HGS | 8.84 | 0.00% | 2031s | 15.00 | 0.00% | 5416s | 21.93 | 0.00% | 13298s |
| | HGS fast | 8.84 | 0.00% | 16s | 15.02 | 0.12% | 55s | 22.03 | 0.43% | 149s |
| | LKH3 | - | - | - | - | - | - | - | - | - |
| | AM greedy | 9.11 | 2.99% | 0.5s | 15.95 | 6.34% | 1.3s | 23.47 | 7.02% | 3.0s |
| | GANCO-AM greedy | **9.04** | **2.17%** | 0.5s | **15.89** | **5.95%** | 1.2s | **23.29** | **6.20%** | 3.0s |
| | AM 10t | 8.98 | 1.56% | 4s | 15.74 | 4.97% | 11s | 23.23 | 5.91% | 28s |
| | GANCO-AM 10t | **8.95** | **1.20%** | 4s | **15.68** | **4.56%** | 11s | **23.02** | **4.93%** | 28s |
| | AM 100t | 8.92 | 0.82% | 44s | 15.58 | 3.86% | 119s | 23.03 | 4.97% | 284s |
| | GANCO-AM 109t | **8.91** | **0.71%** | 43s | **15.52** | **3.49%** | 120s | **22.81** | **4.01%** | 281s |
| Identical | HGS | 5.93 | 0.00% | 1495s | 9.74 | 0.00% | 4341s | 14.46 | 0.00% | 10521s |
| | HGS fast | 5.93 | 0.01% | 13s | 9.74 | 0.04% | 43s | 14.49 | 0.25% | 136s |
| | LKH3 | - | - | - | - | - | - | - | - | - |
| | AM greedy | 6.19 | 4.47% | 0.5s | 10.25 | 5.31% | 1.1s | 15.75 | 8.93% | 3.4s |
| | GANCO-AM greedy | **6.12** | **3.34%** | 0.5s | **10.20** | **4.75%** | 1.1s | **15.47** | **7.03%** | 3.0s |
| | AM 10t | 6.10 | 2.91% | 4s | 10.14 | 4.15% | 11s | 15.55 | 7.59% | 30s |
| | GANCO-AM 10t | **6.06** | **2.29%** | 4s | **10.09** | **3.60%** | 11s | **15.32** | **5.98%** | 29s |
| | AM 100t | 6.05 | 2.10% | 43s | 10.06 | 3.36% | 120s | 15.41 | 6.63% | 301s |
| | GANCO-AM 109t | **6.03** | **1.74%** | 42s | **10.01** | **2.86%** | 119s | **15.20** | **5.18%** | 284s |
| Quadrant | HGS | 5.83 | 0.00% | 1708s | 9.56 | 0.00% | 5049s | 14.05 | 0.00% | 12235s |
| | HGS fast | 5.83 | 0.04% | 14s | 9.57 | 0.10% | 50s | 14.12 | 0.48% | 152s |
| | LKH3 | - | - | - | - | - | - | - | - | - |
| | AM greedy | 6.05 | 3.72% | 0.5s | 10.24 | 7.11% | 1.1s | 15.37 | 9.40% | 3.0s |
| | GANCO-AM greedy | **6.04** | **3.59%** | 0.5s | **10.18** | **6.52%** | 1.2s | **15.14** | **7.76%** | 3.0s |
| | AM 10t | 5.97 | 2.40% | 4s | 10.08 | 5.46% | 10s | 15.13 | 7.71% | 29s |
| | GANCO-AM 10t | **5.97** | **2.34%** | 4s | **10.03** | **4.89%** | 11s | **14.93** | **6.25%** | 28s |
| | AM 100t | 5.93 | 1.67% | 42s | 9.97 | 4.31% | 117s | 14.96 | 6.50% | 284s |
| | GANCO-AM 109t | **5.93** | **1.66%** | 42s | **9.92** | **3.79%** | 117s | **14.77** | **5.12%** | 280s |
| SL | HGS | 4.40 | 0.05% | 1386s | 6.69 | 0.00% | 4601s | 9.52 | 0.00% | 12151s |
| | HGS fast | 4.40 | 0.09% | 13s | 6.69 | 0.09% | 46s | 9.56 | 0.36% | 145s |
| | LKH3 | 4.40 | 0.00% | 1069s | 6.69 | 0.00% | 5630s | 9.54 | 0.18% | 12595s |
| | AM greedy | 4.59 | 4.24% | 0.4s | 7.17 | 7.21% | 1.1s | 10.67 | 12.06% | 3.3s |
| | GANCO-AM greedy | **4.58** | **4.11%** | 0.4s | **7.13** | **6.59%** | 1.1s | **10.43** | **9.51%** | 2.8s |
| | AM 10t | 4.52 | 2.69% | 3s | 7.03 | 5.19% | 10s | 10.45 | 9.73% | 28s |
| | GANCO-AM 10t | **4.52** | **2.68%** | 3s | **6.98** | **4.49%** | 9s | **10.25** | **7.60%** | 26s |
| | AM 100t | **4.48** | **1.89%** | 38s | 6.95 | 3.93% | 109s | 10.30 | 8.13% | 271s |
| | GANCO-AM 109t | 4.48 | 1.90% | 38s | **6.90** | **3.27%** | 107s | **10.12** | **6.30%** | 258s |

# D   AM EXPERIMENTS ON OP

In the Orienteering Problem (OP), the vehicle starts from the depot, visits some of the customers and returns to the depot. The goal is to maximize the total prize collected from the customers with the tour distance shorter than the maximum length. We use the hardest setting reported in Kool et al. (2019), where the prize of node increases with the *distance* to the depot. The base training distribu-

Table 7: Results of AM and GANCO-AM for OP on instances with different distributions.

| Dist. | Method | $n = 20$ | | | $n = 50$ | | | $n = 100$ | | |
|-------|--------|------|------|------|------|------|------|------|------|------|
| | | Obj. | Gap | Time | Obj. | Gap | Time | Obj. | Gap | Time |
| *Base* | Compass | 5.38 | 0.00% | 49s | 16.19 | 0.00% | 167s | 33.18 | 0.00% | 454s |
| | AM greedy | 5.27 | 1.96% | 0.3s | **15.82** | **2.27%** | 0.8s | **32.40** | **2.34%** | 1.7s |
| | GANCO-AM greedy | **5.28** | **1.89%** | 0.3s | 15.77 | 2.57% | 0.8s | 32.30 | 2.64% | 1.7s |
| | AM 10t | 5.31 | 1.25% | 2s | **15.98** | **1.33%** | 6s | **32.74** | **1.32%** | 15s |
| | GANCO-AM 10t | **5.31** | **1.17%** | 2s | 15.95 | 1.48% | 6s | 32.69 | 1.48% | 15s |
| | AM 100t | 5.33 | 0.81% | 27s | **16.06** | **0.81%** | 67s | **32.92** | **0.77%** | 165s |
| | GANCO-AM 100t | **5.34** | **0.71%** | 27s | 16.04 | 0.91% | 67s | 32.89 | 0.86% | 166s |
| *Clustered* | Compass | 6.57 | 0.00% | 75s | 23.14 | 0.00% | 786s | 51.23 | 0.00% | 2173s |
| | AM greedy | 6.40 | 2.50% | 0.3s | 21.48 | 7.17% | 1.0s | 44.19 | 13.74% | 2.3s |
| | GANCO-AM greedy | **6.42** | **2.22%** | 0.3s | **21.73** | **6.10%** | 0.9s | **46.67** | **8.91%** | 2.3s |
| | AM 10t | 6.46 | 1.62% | 3s | 21.91 | 5.33% | 8s | 45.38 | 11.43% | 21s |
| | GANCO-AM 10t | **6.47** | **1.40%** | 3s | **22.15** | **4.29%** | 8s | **47.48** | **7.32%** | 22s |
| | AM 100t | 6.49 | 1.10% | 32s | 22.18 | 4.14% | 87s | 46.16 | 9.90% | 221s |
| | GANCO-AM 100t | **6.51** | **0.90%** | 32s | **22.40** | **3.21%** | 88s | **48.08** | **6.16%** | 227s |
| *Uniform* | Compass | 7.31 | 0.00% | 90s | 22.46 | 0.00% | 304s | 45.97 | 0.00% | 559s |
| | AM greedy | 7.06 | 3.41% | 0.3s | 21.40 | 4.72% | 0.9s | 41.72 | 9.26% | 2.4s |
| | GANCO-AM greedy | **7.11** | **2.74%** | 0.3s | **21.57** | **3.97%** | 1.0s | **43.29** | **5.84%** | 2.4s |
| | AM 10t | 7.15 | 2.20% | 3s | 21.74 | 3.22% | 8s | 42.54 | 7.46% | 23s |
| | GANCO-AM 10t | **7.19** | **1.66%** | 3s | **21.89** | **2.57%** | 8s | **43.94** | **4.43%** | 23s |
| | AM 100t | 7.20 | 1.54% | 33s | 21.95 | 2.30% | 91s | 43.07 | 6.32% | 241s |
| | GANCO-AM 100t | **7.24** | **1.05%** | 33s | **22.06** | **1.80%** | 91s | **44.37** | **3.48%** | 241s |
| *Diagonal* | Compass | 7.57 | 0.00% | 137s | 26.52 | 0.00% | 752s | 55.33 | 0.00% | 634s |
| | AM greedy | 7.28 | 3.84% | 0.3s | 24.60 | 7.24% | 1.0s | 44.61 | 19.39% | 2.5s |
| | GANCO-AM greedy | **7.36** | **2.75%** | 0.3s | **25.26** | **4.78%** | 1.0s | **51.37** | **7.17%** | 2.4s |
| | AM 10t | 7.37 | 2.64% | 3s | 25.15 | 5.17% | 8s | 46.35 | 16.24% | 23s |
| | GANCO-AM 10t | **7.43** | **1.80%** | 3s | **25.62** | **3.41%** | 8s | **51.95** | **6.12%** | 23s |
| | AM 100t | 7.42 | 1.98% | 33s | 25.53 | 3.76% | 91s | 47.31 | 14.49% | 239s |
| | GANCO-AM 100t | **7.48** | **1.24%** | 33s | **25.87** | **2.47%** | 91s | **52.49** | **5.15%** | 240s |
| *Gaussian* | Compass | 6.14 | 0.00% | 64s | 19.52 | 0.00% | 346s | 40.73 | 0.00% | 927s |
| | AM greedy | 5.99 | 2.51% | 0.3s | 18.48 | 5.35% | 1.0s | 35.16 | 13.68% | 2.4s |
| | GANCO-AM greedy | **6.00** | **2.37%** | 0.3s | **18.56** | **4.93%** | 1.0s | **37.31** | **8.40%** | 2.4s |
| | AM 10t | 6.05 | 1.59% | 3s | 18.81 | 3.66% | 8s | 35.96 | 11.71% | 22s |
| | GANCO-AM 10t | **6.06** | **1.43%** | 3s | **18.88** | **3.31%** | 8s | **37.97** | **6.80%** | 23s |
| | AM 100t | 6.08 | 1.09% | 32s | 19.00 | 2.69% | 90s | 36.57 | 10.22% | 230s |
| | GANCO-AM 100t | **6.09** | **0.93%** | 32s | **19.05** | **2.40%** | 90s | **38.44** | **5.65%** | 238s |
| *TSPLIB-S* | Compass | 6.14 | 0.00% | 63s | 20.47 | 0.00% | 323s | 44.98 | 0.00% | 857s |
| | AM greedy | 6.01 | 2.10% | 0.3s | 19.37 | 5.37% | 1.0s | 39.44 | 12.32% | 2.3s |
| | GANCO-AM greedy | **6.02** | **1.95%** | 0.3s | **19.50** | **4.74%** | 0.9s | **41.37** | **8.03%** | 2.4s |
| | AM 10t | 6.06 | 1.38% | 3s | 19.69 | 3.79% | 8s | 40.58 | 9.80% | 23s |
| | GANCO-AM 10t | **6.07** | **1.22%** | 3s | **19.80** | **3.29%** | 9s | **42.29** | **5.99%** | 23s |
| | AM 100t | 6.09 | 0.89% | 32s | 19.88 | 2.90% | 90s | 41.24 | 8.32% | 232s |
| | GANCO-AM 100t | **6.09** | **0.79%** | 33s | **19.97** | **2.43%** | 92s | **42.79** | **4.87%** | 238s |

tion is the unit uniform distribution for the node coordinate. The maximum tour length is fixed as 2, 3 and 4 for OP with 20, 50 and 100 customers, respectively. The prize $\rho_i$ of node $i$ is determined by $\rho_i = \frac{1}{100}\lfloor 1 + 99 \times \frac{w_{0i}}{max_{j=1}^{n} w_{0j}} \rfloor$, where $w_{0i}$ is the distance between depot and node $i$. Therefore, the generation model only learns the node coordinates. For the generalization testing distributions, we use the same coordinate distributions for TSP (*Clustered*, *Uniform*, *Diagonal*, *Gaussian* and *TSPLIB-S*) as the ones for the customers. In each generalization distribution, the depot coordinates follow the 3 depot positioning (*Central*, *Eccentric*, *Random*) with equal probabilities.

In Table 7, we report the results of Compass Kobeaga et al. (2018), AM and GANCO-AM on the testing distributions for OP. Different from most other routing problems, the objective value is the total collected prize which is to be maximized instead of minimized. The results confirm that the GANCO framework significantly improves the performance on all the generalization settings with different testing distributions, graph sizes and numbers of solution trajectories.

# E   AM EXPERIMENTS ON PCTSP

In Prize Collecting TSP (PCTSP), the vehicle starts from the depot, visits some of the customers to collect prize and returns to the depot. With the total collected prize no less than a minimum value, the goal is to minimize the tour distance plus the total penalties of unvisited nodes. In Kool et al. (2019), the base training distribution is that the coordinates of depot and customers follow the unit uniform distribution. The prizes are designed to force the vehicle to visit approximately

Table 8: Results of AM and GANCO-AM for PCTSP on instances with different distributions.

| Dist. | Method | $n = 20$ | | | $n = 50$ | | | $n = 100$ | | |
| | | Obj. | Gap | Time | Obj. | Gap | Time | Obj. | Gap | Time |
|---|---|---|---|---|---|---|---|---|---|---|
| Base | ILS | 3.15 | 0.00% | 862s | 4.50 | 0.00% | 6895s | 5.98 | 0.00% | 47404s |
| | ILS fast | 3.19 | 1.10% | 78s | 4.57 | 1.56% | 119s | 6.10 | 2.12% | 358s |
| | AM greedy | 3.15 | 0.11% | 0.3s | 4.54 | 0.94% | 0.7s | 6.11 | 2.15% | 1.7s |
| | GANCO-AM greedy | **3.15** | **0.09%** | 0.3s | **4.54** | **0.90%** | 0.7s | **6.10** | **2.09%** | 1.7s |
| | AM 10t | 3.15 | -0.19% | 3s | **4.51** | **0.36%** | 7s | 6.06 | 1.33% | 16s |
| | GANCO-AM 10t | **3.14** | **-0.20%** | 3s | 4.51 | 0.38% | 7s | **6.05** | **1.29%** | 16s |
| | AM 100t | **3.14** | **-0.37%** | 32s | **4.50** | **0.08%** | 74s | 6.03 | 0.85% | 168s |
| | GANCO-AM 100t | 3.14 | -0.36% | 33s | 4.50 | 0.10% | 75s | **6.03** | **0.83%** | 174s |
| Clustered | ILS | 2.90 | 0.00% | 1018s | 3.72 | 0.00% | 9957s | 4.65 | 0.00% | 57305s |
| | ILS fast | 2.93 | 1.04% | 81s | 3.79 | 1.94% | 102s | 4.77 | 2.74% | 476s |
| | AM greedy | 2.90 | 0.14% | 0.3s | 3.79 | 2.06% | 0.9s | 5.09 | 9.53% | 2.2s |
| | GANCO-AM greedy | **2.90** | **0.01%** | 0.3s | **3.76** | **1.21%** | 0.8s | **4.89** | **5.14%** | 2.1s |
| | AM 10t | 2.89 | -0.28% | 3s | 3.75 | 0.95% | 8s | 4.99 | 7.40% | 21s |
| | GANCO-AM 10t | **2.89** | **-0.35%** | 3s | **3.73** | **0.37%** | 8s | **4.82** | **3.78%** | 21s |
| | AM 100t | 2.88 | -0.50% | 34s | 3.73 | 0.29% | 88s | 4.92 | 5.95% | 224s |
| | GANCO-AM 100t | **2.88** | **-0.54%** | 34s | **3.71** | **-0.13%** | 88s | **4.78** | **2.76%** | 218s |
| Uniform | ILS | 2.68 | 0.00% | 1121s | 3.63 | 0.00% | 12893s | 4.72 | 0.00% | 63008s |
| | ILS fast | 2.71 | 1.14% | 76s | 3.69 | 1.64% | 101s | 4.83 | 2.16% | 523s |
| | AM greedy | 2.69 | 0.43% | 0.3s | 3.75 | 3.33% | 0.9s | 5.23 | 10.70% | 2.3s |
| | GANCO-AM greedy | **2.69** | **0.38%** | 0.3s | **3.70** | **1.94%** | 0.9s | **4.99** | **5.74%** | 2.3s |
| | AM 10t | 2.68 | -0.03% | 3s | 3.70 | 2.00% | 8s | 5.13 | 8.65% | 22s |
| | GANCO-AM 10t | **2.68** | **-0.04%** | 3s | **3.67** | **1.13%** | 8s | **4.94** | **4.51%** | 22s |
| | AM 100t | 2.67 | -0.26% | 34s | 3.68 | 1.23% | 92s | 5.06 | 7.18% | 235s |
| | GANCO-AM 100t | **2.67** | **-0.26%** | 34s | **3.65** | **0.64%** | 91s | **4.90** | **3.65%** | 231s |
| Diagonal | ILS | 2.49 | 0.00% | 1242s | 2.95 | 0.00% | 17265s | 3.44 | 0.00% | 83044s |
| | ILS fast | 2.52 | 1.24% | 70s | 3.01 | 2.11% | 122s | 3.51 | 2.13% | 716s |
| | AM greedy | 2.51 | 0.73% | 0.3s | 3.23 | 9.69% | 0.9s | 4.62 | 34.47% | 2.3s |
| | GANCO-AM greedy | **2.50** | **0.37%** | 0.3s | **3.09** | **4.73%** | 0.9s | **3.97** | **15.53%** | 2.3s |
| | AM 10t | 2.49 | 0.04% | 3s | 3.12 | 6.06% | 8s | 4.46 | 29.68% | 23s |
| | GANCO-AM 10t | **2.48** | **-0.16%** | 3s | **3.04** | **3.09%** | 8s | **3.89** | **13.32%** | 22s |
| | AM 100t | 2.48 | -0.28% | 35s | 3.07 | 4.10% | 92s | 4.32 | 25.68% | 234s |
| | GANCO-AM 100t | **2.48** | **-0.41%** | 35s | **3.01** | **2.04%** | 91s | **3.82** | **11.29%** | 229s |
| Gaussian | ILS | 2.80 | 0.00% | 949s | 3.59 | 0.00% | 11168s | 4.48 | 0.00% | 36622s |
| | ILS fast | 2.82 | 0.83% | 71s | 3.63 | 1.06% | 99s | 4.53 | 1.17% | 505s |
| | AM greedy | 2.81 | 0.31% | 0.3s | 3.68 | 2.69% | 0.8s | 4.94 | 10.21% | 2.2s |
| | GANCO-AM greedy | **2.81** | **0.31%** | 0.3s | **3.66** | **1.97%** | 0.8s | **4.74** | **5.76%** | 2.1s |
| | AM 10t | **2.80** | **-0.03%** | 3s | 3.65 | 1.60% | 8s | 4.84 | 8.06% | 22s |
| | GANCO-AM 10t | 2.80 | -0.02% | 3s | **3.63** | **1.19%** | 8s | **4.68** | **4.37%** | 21s |
| | AM 100t | **2.79** | **-0.21%** | 34s | 3.62 | 1.00% | 88s | 4.78 | 6.66% | 223s |
| | GANCO-AM 100t | 2.80 | -0.19% | 34s | **3.62** | **0.75%** | 86s | **4.64** | **3.45%** | 215s |
| TSPLIB-S | ILS | 2.98 | 0.00% | 952s | 3.93 | 0.00% | 11155s | 4.96 | 0.00% | 39783s |
| | ILS fast | 3.01 | 0.99% | 72s | 3.99 | 1.62% | 106s | 5.07 | 2.25% | 471s |
| | AM greedy | 2.99 | 0.15% | 0.3s | 4.00 | 1.81% | 0.9s | 5.40 | 8.83% | 2.4s |
| | GANCO-AM greedy | **2.99** | **0.08%** | 0.3s | **3.98** | **1.43%** | 0.9s | **5.18** | **4.51%** | 2.2s |
| | AM 10t | 2.98 | -0.20% | 3s | 3.97 | 1.01% | 8s | 5.31 | 7.00% | 23s |
| | GANCO-AM 10t | **2.98** | **-0.24%** | 3s | **3.96** | **0.75%** | 8s | **5.12** | **3.30%** | 22s |
| | AM 100t | 2.97 | -0.39% | 35s | 3.95 | 0.56% | 89s | 5.24 | 5.68% | 237s |
| | GANCO-AM 100t | **2.97** | **-0.40%** | 35s | **3.94** | **0.37%** | 90s | **5.09** | **2.51%** | 221s |

half of the customer nodes, which follows the uniform distribution $U(0, 4/n)$. And the penalties are designed to balance the importance of minimizing the tour distance and the cost of unvisited nodes, which follows the uniform distribution $U(0, 3K/n)$. Particularly, K is fixed as 2, 3 and 4 for PCTSP with 20, 50 and 100 customers. The prizes and penalties are specifically designed to have meaningful instances. Therefore, for the generalization testing distributions, we keep the node prize and penalty distributions the same as the base training distribution. We only consider the generalization to different node coordinate distributions. The depot coordinates follow one of the three depot positioning (*Central*, *Eccentric*, *Random*) with equal probabilities. The customer coordinates follow the same generalization testing distributions as those for TSP.

In Table 8, we show the results of ILS, a faster version of ILS, AM and GANCO-AM on PCTSP with different instance distributions. A faster version of ILS (ILS fast) uses the number of non-improvement iterations, the maximum iterations and the maximum reboot times as 200, 400 and 201 instead of the default 20000, 40000 and 4001 for the original ILS. The ILS fast is the non-learning baseline algorithm to train the generation model. Similarly, the GANCO framework improves the original AM by large margins on almost all the generalization testing settings. For the only exception (PCTSP20 with *Gaussian*), the performance of AM and GANCO-AM is almost identical. It is

Table 9: Results of AM and GANCO-AM for KP on instances with different distributions.

| Dist. | Method | n = 50 | | | n = 100 | | | n = 200 | | |
|---|---|---|---|---|---|---|---|---|---|---|
| | | Obj. | Gap | Time | Obj. | Gap | Time | Obj. | Gap | Time |
| *Base* | DP | 20.099 | 0.00% | 23s | 40.404 | 0.00% | 91s | 57.730 | 0.00% | 186s |
| | AM greedy | **20.081** | **0.092%** | 0.7s | **40.383** | **0.052%** | 1.7s | **57.699** | **0.053%** | 4.1s |
| | GANCO-AM greedy | 20.080 | 0.096% | 0.7s | 40.381 | 0.056% | 1.7s | 57.697 | 0.058% | 3.9s |
| | AM 10t | **20.088** | **0.053%** | 6s | **40.393** | **0.028%** | 16s | **57.714** | **0.028%** | 37s |
| | GANCO-AM 10t | 20.088 | 0.056% | 6s | 40.392 | 0.029% | 16s | 57.712 | 0.031% | 38s |
| | AM 100t | **20.092** | **0.036%** | 67s | 40.397 | 0.018% | 169s | **57.721** | **0.015%** | 391s |
| | GANCO-AM 100t | 20.091 | 0.039% | 68s | **40.397** | **0.018%** | 173s | 57.719 | 0.019% | 404s |
| *Clustered* | DP | 18.915 | 0.00% | 22s | 38.084 | 0.00% | 97s | 53.754 | 0.00% | 182s |
| | AM greedy | 18.873 | 0.226% | 0.8s | 38.018 | 0.174% | 2.1s | 53.544 | 0.390% | 6.4s |
| | GANCO-AM greedy | **18.880** | **0.186%** | 0.8s | **38.029** | **0.144%** | 2.1s | **53.654** | **0.184%** | 6.4s |
| | AM 10t | 18.891 | 0.127% | 8s | 38.048 | 0.094% | 21s | 53.601 | 0.283% | 63s |
| | GANCO-AM 10t | **18.895** | **0.110%** | 8s | **38.052** | **0.084%** | 21s | **53.694** | **0.111%** | 64s |
| | AM 100t | 18.900 | 0.080% | 81s | 38.062 | 0.057% | 241s | 53.637 | 0.217% | 677s |
| | GANCO-AM 100t | **18.901** | **0.074%** | 81s | **38.064** | **0.053%** | 239s | **53.716** | **0.071%** | 717s |
| *Uniform* | DP | 18.783 | 0.00% | 22s | 37.757 | 0.00% | 95s | 50.819 | 0.00% | 200s |
| | AM greedy | 18.741 | 0.223% | 0.7s | 37.695 | 0.163% | 1.8s | 50.731 | 0.174% | 4.1s |
| | GANCO-AM greedy | **18.753** | **0.159%** | 0.7s | **37.717** | **0.105%** | 1.8s | **50.748** | **0.140%** | 4.1s |
| | AM 10t | 18.757 | 0.137% | 6s | 37.720 | 0.095% | 18s | 50.776 | 0.086% | 41s |
| | GANCO-AM 10t | **18.764** | **0.097%** | 6s | **37.732** | **0.065%** | 18s | **50.783** | **0.073%** | 41s |
| | AM 100t | 18.765 | 0.094% | 71s | 37.733 | 0.061% | 182s | 50.795 | 0.048% | 443s |
| | GANCO-AM 100t | **18.769** | **0.071%** | 71s | **37.740** | **0.044%** | 182s | **50.798** | **0.042%** | 443s |
| *Diagonal* | DP | 18.650 | 0.00% | 25s | 37.031 | 0.00% | 90s | 49.553 | 0.00% | 194s |
| | AM greedy | 18.586 | 0.344% | 0.7s | 36.956 | 0.202% | 1.8s | 49.419 | 0.271% | 4.2s |
| | GANCO-AM greedy | **18.620** | **0.158%** | 0.7s | **36.980** | **0.138%** | 1.8s | **49.436** | **0.236%** | 4.2s |
| | AM 10t | 18.612 | 0.202% | 6s | 36.990 | 0.111% | 18s | 49.477 | 0.154% | 42s |
| | GANCO-AM 10t | **18.634** | **0.084%** | 6s | **37.002** | **0.078%** | 18s | **49.481** | **0.145%** | 42s |
| | AM 100t | 18.625 | 0.135% | 71s | 37.006 | 0.068% | 182s | **49.508** | **0.091%** | 445s |
| | GANCO-AM 100t | **18.639** | **0.057%** | 71s | **37.012** | **0.053%** | 182s | 49.506 | 0.096% | 444s |
| *Gaussian* | DP | 16.351 | 0.00% | 22s | 32.126 | 0.00% | 97s | 37.127 | 0.00% | 193s |
| | AM greedy | 16.319 | 0.196% | 0.8s | 32.088 | 0.120% | 1.9s | 37.057 | 0.191% | 3.8s |
| | GANCO-AM greedy | **16.319** | **0.193%** | 0.7s | **32.089** | **0.116%** | 2.0s | **37.074** | **0.143%** | 3.8s |
| | AM 10t | **16.334** | **0.103%** | 7s | **32.108** | **0.057%** | 19s | 37.096 | 0.086% | 38s |
| | GANCO-AM 10t | 16.333 | 0.107% | 7s | 32.107 | 0.058% | 19s | **37.100** | **0.074%** | 38s |
| | AM 100t | **16.341** | **0.060%** | 75s | **32.116** | **0.030%** | 205s | 37.109 | 0.049% | 402s |
| | GANCO-AM 100t | 16.339 | 0.070% | 75s | 32.116 | 0.032% | 203s | **37.113** | **0.038%** | 403s |
| *TSPLIB-S* | DP | 18.917 | 0.00% | 22s | 37.850 | 0.00% | 91s | 51.699 | 0.00% | 189s |
| | AM greedy | 18.887 | 0.159% | 0.8s | 37.799 | 0.134% | 2.0s | 51.579 | 0.233% | 4.9s |
| | GANCO-AM greedy | **18.889** | **0.148%** | 0.8s | **37.810** | **0.105%** | 2.0s | **51.621** | **0.151%** | 5.0s |
| | AM 10t | **18.900** | **0.089%** | 7s | 37.821 | 0.075% | 19s | 51.626 | 0.142% | 49s |
| | GANCO-AM 10t | 18.899 | 0.092% | 7s | **37.827** | **0.059%** | 19s | **51.658** | **0.080%** | 49s |
| | AM 100t | **18.906** | **0.058%** | 75s | 37.831 | 0.048% | 203s | 51.647 | 0.102% | 493s |
| | GANCO-AM 100t | 18.904 | 0.066% | 75s | **37.834** | **0.040%** | 211s | **51.674** | **0.049%** | 498s |

worth noting that for small size problem, AM and GANCO-AM can outperform the traditional non-learning methods that are highly optimized for the specific problems.

# F  AM EXPERIMENTS ON KP

Following Kwon et al. (2020), the item weights and values in KP are sampled from the unit uniform distribution $U(0, 1)$ as the base training distribution. The weight capacity is fixed as 12.5, 25, 25 for the instances with 50, 100, 200 items. Considering the weights and values as the $x$ and $y$ axis coordinates, we could leverage the same generalization testing distributions for TSP. The item weights are rounded to 2 decimal places for the Dynamic Programming (DP) algorithm to attain the optimal solutions.

As shown in Table 9, for the greedy inferring, the GANCO framework effectively improves the performance on all the generalization testing distributions for different problem sizes. With more trajectories inferred, the performance of AM and GANCO-AM become fairly close on some distributions (such as *Gaussian* and *TSPLIB-S*) because KP is a relatively simple problem and the order of node (item) sequences will not affect the solution. Therefore, multiple trajectories by sampling introduce more randomness to the solution quality and the performance slightly fluctuates.

## G   POMO EXPERIMENTS

We leverage AM as the base model for most experiments given that it has been used in many other recent works including POMO, and exhibits good performance on a series of COPs. Here, we only consider TSP100 and CVRP100 since the results of the improved AM in our experiments are superior or close to those of POMO on the other sizes. Specifically, regarding TSP20 and TSP50, the gaps of AM are 0.14% and 0.73% compared to 0.12% and 0.64% for POMO. Regarding CVRP20 and CVRP50, the gaps of AM are 2.74% and 4.05% compared to 3.72% and 3.52% for POMO. Regarding KP50, KP100 and KP200, the gaps of AM are 0.092%, 0.052% and 0.053% compared to 0.130%, 0.125% and 0.260% for POMO. However, POMO significantly outperforms AM on TSP100 and CVRP100 (2.16% and 5.22% of AM for TSP100 and CVRP100 compared to 1.07% and 3.32%). Therefore, to further show that the proposed GANCO framework is model agnostic and general enough to other models, we conduct experiments on POMO to solve TSP100 and CVRP100.

The results of POMO and GANCO-POMO are shown in Table 10 and Table 3 for TSP100 and CVRP100, respectively. Both the results of single trajectory and 100 trajectories (the-multi-starting-node strategy) are presented. Clearly, the improvement of GANCO-POMO over POMO is salient for both decoding methods.

Table 10: Results of POMO and GANCO-POMO for TSP100.

| Dist. | Concorde | | Method | single trajectory | | | 100 trajectories | | |
|---|---|---|---|---|---|---|---|---|---|
| | Obj. | Time | | Obj. | Gap | Time | Obj. | Gap | Time |
| *Base* | 7.76 | 142s | POMO | **7.84** | **1.07%** | 2.2s | **7.79** | **0.45%** | 34s |
| | | | GANCO-POMO | 7.85 | 1.20% | | 7.80 | 0.54% | |
| *Clustered* | 5.15 | 210s | POMO | 5.42 | 5.26% | 2.2s | 5.26 | 2.29% | 34s |
| | | | GANCO-POMO | **5.28** | **2.61%** | | **5.21** | **1.15%** | |
| *Uniform* | 5.30 | 183s | POMO | 5.56 | 4.97% | 2.2s | 5.41 | 2.21% | 34s |
| | | | GANCO-POMO | **5.39** | **1.82%** | | **5.34** | **0.75%** | |
| *Diagonal* | 3.24 | 196s | POMO | 3.94 | 21.78% | 2.2s | 3.57 | 10.43% | 34s |
| | | | GANCO-POMO | **3.30** | **2.12%** | | **3.26** | **0.86%** | |
| *Gaussian* | 5.70 | 167s | POMO | 5.93 | 3.99% | 2.2s | 5.80 | 1.84% | 34s |
| | | | GANCO-POMO | **5.82** | **2.07%** | | **5.75** | **0.97%** | |
| *TSPLIB-S* | 5.76 | 155s | POMO | 5.93 | 2.97% | 2.2s | 5.84 | 1.42% | 34s |
| | | | GANCO-POMO | **5.86** | **1.79%** | | **5.80** | **0.74%** | |

## H   MODIFICATIONS TO IMPROVE AM

We modify the base AM to improve its performance in two ways. Firstly, we remove the Batch Normalization layers as we find that they only help the model converge slightly faster at the beginning but harm the generalization performance on other distributions. Secondly, we train the model until full convergence instead of the fixed 100 epochs used in Kool et al. (2019). Therefore, the performance of the base model is much better than the original one (e.g., 0.14%, 0.73%, 2.16% optimality gaps instead of the original 0.34%, 1.76%, 4.53% for TSP with 20, 50, 100 nodes).

In Table 11, we take TSP50 as an example and show the results of AM with and without the BN layers. As the results are reported for 100 epochs in Kool et al. (2019), we start from there and present the results every 300 epochs until convergence. Clearly, after convergence, the performance without BN is better than with on various instance distributions including the training one. And the performance on most generalization distributions gets better with more training epochs. Even though the performance on some distributions could fluctuate or deteriorate, the best results on these distributions among different number of epochs are still far inferior to the ones attained after GANCO training.

To be more fair, we further compare GANCO-AM against a variant of AM named AM-S, which is trained on the base distribution for the same number of steps (also the same number of instances) after the pretraining stage as GANCO-AM. Taking TSP50 as an example, the average performance of AM (after the pretraining stage), AM-S and GANCO-AM on the five generalization distributions is 4.156, 4.168 and 3.989, respectively (the smaller the better). The generalization ability does not improve with more epochs for the original AM. And the improvement of GANCO over the other two models is clearly significant. The detailed results are shown in Table 11.

Table 11: Results of AM with and without the Batch Normalization (BN) layers trained for different number of epochs on TSP50.

| Dist. | Models | 100 epochs | 400 epochs | 700 epochs | 1000 epochs | 1300 epochs | AM-S (without BN) | GANCO-AM (without BN) |
|---|---|---|---|---|---|---|---|---|
| Base | AM with BN | 5.806 | 5.753 | 5.740 | 5.735 | 5.731 | - | 5.735 |
| | AM without BN | 5.825 | 5.755 | 5.740 | 5.733 | **5.729** | 5.729 | |
| Clustered | AM with BN | 4.235 | 4.191 | 4.171 | 4.183 | 4.181 | - | 4.096 |
| | AM without BN | 4.295 | 4.153 | 4.142 | 4.142 | **4.137** | 4.145 | |
| Uniform | AM with BN | 4.214 | 4.080 | 4.219 | 4.159 | 4.226 | - | 3.970 |
| | AM without BN | 4.117 | 4.057 | 4.044 | 4.089 | **4.045** | 4.071 | |
| Diagonal | AM with BN | 3.292 | 3.205 | 3.886 | 4.288 | 4.411 | - | 2.784 |
| | AM without BN | 3.858 | 3.260 | 3.427 | 3.457 | **3.351** | 3.368 | |
| Gaussian | AM with BN | 4.671 | 4.621 | 4.667 | 4.685 | 4.687 | - | 4.542 |
| | AM without BN | 4.743 | 4.639 | 4.659 | 4.691 | **4.671** | 4.684 | |
| TSPLIB-S | AM with BN | 4.656 | 4.605 | 4.606 | 4.606 | 4.592 | - | 4.554 |
| | AM without BN | 4.656 | 4.600 | 4.582 | 4.578 | **4.577** | 4.572 | |

## I  MORE ANALYSIS ON GANCO FRAMEWORK

Table 12: Results of AM, AM-T4 and GANCO-AM for TSP100 and OP100. Distributions with '*' are the AM-T4 training distributions.

| Dist. | TSP100 | | | OP100 | | |
|---|---|---|---|---|---|---|
| | AM | AM-T4 | GANCO-AM | AM | AM-T4 | GANCO-AM |
| Base* | 7.93 | 7.97 | 7.94 | 32.40 | 32.22 | 32.30 |
| Clustered | 5.60 | 5.57 | **5.46** | 44.19 | 45.55 | **46.67** |
| Uniform* | 5.67 | 5.52 | 5.51 | 41.72 | 43.65 | 43.29 |
| Diagonal* | 5.22 | 3.50 | 3.47 | 44.61 | 52.76 | 51.37 |
| Gaussian* | 6.21 | 6.27 | 6.10 | 35.16 | 36.25 | 37.31 |
| TSPLIB-S | 6.06 | 6.02 | **5.99** | 39.44 | 40.26 | **41.37** |

We conduct experiments of training the original AM on 4 distributions (*base*, *uniform*, *diagonal*, *gaussian*) after the pre-training stage, namely, AM-T4. The training dataset equally samples from the 4 distributions and AM-T4 is trained for the same number of steps (also the same number of instances) as the GANCO adversarial training stage. For TSP100, the average performance of the original AM (after the pretraining stage), AM-T4 and GANCO-AM is 6.26, 5.82 and 5.75 on the four AM-T4 training distributions and 5.83, 5.79 and 5.73 on the other two generalization distributions (the smaller the better). For OP100, the average performance of the original AM, AM-T4 and GANCO-AM is 38.47, 41.22 and 41.07 on the four AM-T4 training distributions and 41.82, 42.90 and 44.02 on the other two generalization distributions (larger collected prizes are better). The detailed results are shown in Table 12.

On the four distributions AM-T4 is trained on, GANCO-AM achieves comparable or better performance than AM-T4. The performance of GANCO-AM is better than AM-T4 on some distributions even though AM-T4 trains on these distributions. And AM-T4 underperforms the original AM on one AM-T4 training distribution (*gaussian*) for TSP100. This is because equally sampling from four distributions leads to unbalanced reward (the rewards for instances with some distributions dominate the gradients). Instead of equally sampling from each of the four distributions, we also tried another simple composition rule (1/2 *base* distribution and 1/6 for each of the three generalization distributions) and the results are similar. In contrast, our GANCO framework uses the alternative training process to dynamically find the hard distributions and adjust the proportion of newly generated distributions in the training dataset during the adversarial training stage. Compared to AM-T4, GANCO-AM achieves most of the benefits of training on in-distribution data.

More importantly, on the two generalization distributions which both models do not train on, the average performance of GANCO-AM is much better, which also demonstrates the improvement of generalization ability by training on adversarially generated distributions.

To demonstrate the adaptive nature of GANCO's adversarial instance generation, we also conduct the experiments of training the agent for one problem (OP) on the distribution curriculum generated by GANCO for another agent for a different problem (PCTSP), namely, GANCO-AM-DIFF. The average performance of GANCO-AM-DIFF on the five generalization distributions is 43.26, 43.55 and 43.90 with 5, 10 and 20 adversarial iterations (the larger the better for OP). In comparison, results of the original GANCO-AM for OP are 43.41, 43.73 and 44.00, respectively. Result of the original AM is 41.02. We can see that the performance of GANCO-AM-DIFF is also pretty good as these two problems are very similar with the distinction of flipping the constraint and objective value. However, the original GANCO-AM for OP performs better, suggesting that the generation model indeed tailors distributions bespoke to the specific problem.

