# OpenReview forum: "Generative Adversarial Training for Neural Combinatorial Optimization Models"
_ICLR.cc/2022/Conference — ICLR 2022 Submitted_

### Official Review · Reviewer_mumN · 2021-10-26

**Correctness:** 2
**Technical Novelty And Significance:** 3
**Empirical Novelty And Significance:** 2
**Recommendation:** 3
**Confidence:** 4

**Main Review:**

## Strengths
1. The idea of generative adversarial training for combinatorial optimization is new.
1. The experiment results are comprehensive and detailed, mainly on various settings of routing problems.
1. Figure 1 presents a clear and straightforward overview of the proposed framework.

## Weaknesses
1. My first concern about this paper is the motivation and necessity for developing generative adversarial training methods for combinatorial optimization problems. Since we are handling a reinforcement learning problem without any supervision labels, it seems not very difficult to obtain a batch of problem instances that lie within the distribution of test instances (for example, loading the business data from last month and using them for training, and predicting on the next month). More aggressively, you may also train (or more properly, finetune) a neural network on the test instance if the time budget is acceptable. Besides, the results on the "base" distribution entry show that the performance of the model will probably degenerate after generative adversarial learning, but such property seems undesirable, especially we can easily obtain a batch of test instances in real-world applications.
1. An assumption made in this paper is that the neural network has enough capacity to learn the input-to-optimal mapping of combinatorial problems from various distributions. If the network does not have enough capacity, the expressiveness of the neural network should be our major concern instead of the generalization ability. However, the results of neural networks, whether aided with adversarial learning or not, are all inferior to the traditional solvers. It makes me doubt the capacity of the current neural network models.
2. Besides, is there any special design to prevent the neural network from "forgetting" the original distribution? The results on the "base" distribution entry show that the performance of the model will probably degenerate after generative adversarial learning, suggesting that the model seems incapable to learn the input-to-optimal mapping under the current pipeline and is likely to forget some information from the original distribution.
1. Concerning the experiment part, it will be very interesting to see the result of traditional algorithms that are configured to run comparatively faster with learning-based methods. For example, the number of trials of LKH-3 is configurable to control the runtime of the heuristic and the current configuration (10000 trials as written in the appendix) seems too time-consuming. On page 7 the authors mentioned that "Generally, their (the neural network's) run time is much shorter than the traditional algorithm as indicated in the table(s), which is a core strength of deep models." I doubt whether such a conclusion still holds if you replace your current solvers with a well-configured LKH-3 solver. It will make much more sense if you do not set a large number of trials for LKH-3 to force it to run super slow.
1. The authors make an effort on including the knapsack problem. However, there is still room for further improvement if the authors can include more problems beyond routing problems because the title of this paper is claimed for general combinatorial optimization problems.
1. The writing and formatting of this paper need to be improved to meet the quality of ICLR papers. See some examples in "detailed comments" below.

## Other detailed comments:
1. It seems unclear whether the objective function $O_\Phi(x)$ is to be maximized or minimized. Maximizing or minimizing $O_\Phi(x)$ will correspond to different formulations of $G(x,\Phi, B)$.
1. In Table 1, the inference times of AM and GANCO-AM are regarded as the same, which is reasonable because they share the same model architecture. And in Table 2 and 3, the inference times of AM/GANCO-AM and POMO/GANCO-POMO are different, which is also reasonable due to random noises in experiments. Although both of the writings may be reasonable, I think it will be better to maintain a consistent style of tables in one paper.
2. In Table 2, the inference time of HGS (~0.5 hours for the smallest sized problem) seems unacceptable as an intermediate step when computing the reward signal in reinforcement learning.
1. What is the time cost for the pertaining stage and the proposed adversarial training steps?


**Summary Of The Paper:**

This paper presents a generative adversarial training pipeline for general combinatorial optimization problems, aiming to improve the generalization ability of learned neural network solvers on unseen data distributions. In experiments, the authors select the attention model as the neural network solver, and REINFORCE, POMO as the RL algorithms. The experiment results are comprehensive on various routing problems, and on the knapsack problem.

**Summary Of The Review:**

The generative adversarial training for combinatorial optimization in this paper is novel, the experiment results are detailed and comprehensive. However, the motivation, writing, and the experiment part of this paper seem to have room for further improvement, and I am suggesting a rejection for this time and I am wishing to see an improved version in the future.

---

> ### Author Response · Authors · 2021-11-17
> **Response (2/2)**
>
> 4. (Continued) With the potential of deep learning models, we study to improve the generalization ability of deep learning models, which could also apply to more powerful models in the future. And the temporary inferiority of deep NCO models compared to some highly optimized traditional algorithms does not contradict the necessity of studying deep NCO models (as in other dozens of related works on deep NCO models such as AM and POMO).
>
> 5.  The proposed GANCO framework is model agnostic and generally applicable to various NCO models for solving different COPs. We use vehicle routing problems as they are one of the most important families of COPs, and deep NCO models for these problems are relatively well developed. Compared to POMO [Kwon2020], which is another deep learning work to solve COPs, we include all three problems in their work and test on two more other routing problems. We will conduct experiments on other COPs and try to include the results in the final version.
>
> 6. Please refer to our responses for the detailed comments.
>
> For the other detailed comments,
>
> 1. We follow the convention in many literature (such as [Blum2003]) to formulate the combinatorial optimization problems as minimizing the objective functions. Without loss of generality, the maximization problem could be viewed as minimizing the negative objective function. We have stated it more clearly in the revised version of our paper.
>
> 2. Thank you for noticing that. We intentionally make Table 1 different from Table 2 and 3 to be more precise. The different inference times in Table 2 and 3 are due to the different numbers of construction steps instead of randomness. With the same model architecture, the inferring time is only related to the number of construction steps (i.e., the number of nodes visited for the routing problems). For TSP in Table 1, the numbers of construction steps are guaranteed to be the same as each tour will have exactly n+1 nodes (the vehicle will visit each node once and return to the starting node). But for CVRP in Table 2 and 3, the depot could be visited for different times and the number of construction steps could be different for the same instance. Therefore, different models with the same architecture will have different inferring times. We have revised Table 1 and Table 4 to be consistent.
>
> 3. The result of HGS in Table 2 uses the recommended number of non-improvement iterations before termination as 20000. As we have mentioned in the 2nd paragraph of Section 4, we use a fast version of HGS for the adversarial training: “As our GANCO framework is relatively robust to the performance of (non-learning) baseline algorithms, we set the number of non-improvement iterations before termination for HGS as 100 instead of the default 20000 to save training time.” The results of the fast HGS (the baseline version) is listed in Table 6 (with the name HGS fast). Please refer to the details in the 2nd paragraph in Section 4 and the 2nd paragraph in Appendix C.
>
> 4.  The longest adversarial training time of our method (GANCO-AM for CVRP100) costs less than 3 days compared to the 14 days for pretraining AM. Please refer to the 3rd paragraph in Section 4.5.
>
> [Kool2019] Wouter Kool, Herke van Hoof, and Max Welling.  Attention, learn to solve routing problems!   In International Conference on Learning Representations, 2019.
> [Xin2021] Liang Xin, Wen Song, Zhiguang Cao, Jie Zhang.  NeuroLKH: Combining Deep Learning Model with Lin-Kernighan-Helsgaun Heuristic for Solving the Traveling Salesman Problem. arXiv preprint arXiv:2110.07983, 2021.
> [Kwon2020] Yeong-Dae Kwon, Jinho Choo, Byoungjip Kim, Iljoo Yoon, Youngjune Gwon, and Seungjai Min. Pomo: Policy optimization with multiple optima for reinforcement learning. Advances in Neural Information Processing Systems, 33, 2020.
> [Blum2003] Christian Blum, and Roli Andrea. Metaheuristics in combinatorial optimization: Overview and conceptual comparison[J]. ACM computing surveys (CSUR), 2003, 35(3): 268-308.

---

> > ### Comment · Reviewer_mumN · 2021-11-22
> > **Thank the authors for the response and here are my further concerns**
> >
> > Dear authors, AC, and other reviewers,
> >
> > I would thank the authors for the detailed response, but I still have the following concerns, and I am not convinced that this paper should be accepted with the current version. In the following I list my further concerns:
> >
> > 1. About the performance drop in the training distribution, the authors mentioned in their rebuttal that "only a 5.56% drop on the base distribution". However, please note that in combinatorial optimization, a performance drop of 5.56% should not be described by "only". It means losing 5.56% of the revenue for your company. Please note that in combinatorial optimization, optimality is always our pursuit, thus the performance drop is never a minor issue. To resolve this issue, I think it may correspond to my second concern, the model capacity.
> >
> > 2. **The authors are studying the overfitting issue for underfitting models**. As I have mentioned in my review, it seems that the current model capacity is not large enough to learn from various data distributions. Since the state-of-the-art model does not have enough capacity to learn from various data distributions, I think we are not at the stage to care about the generalization ability. One may argue that the model capacity will definitely grow, and sooner or later we will have our killer architectures for CO like ResNet/Transformers with large model capacity and are easy to train. Yes, I am also optimistic about this, but the approaches are developed and tested for the current model architectures (like AM), and I would like to emphasize that these methods may not work for those models with enough capacity in the future. I would like to point out that the authors claim they are solving the issue of overfitting, but as far as I know, the current state-of-the-art models are suffering from underfitting. Generalization study should be valuable only if the models have enough capacity and are truly suffering from overfitting.
> >
> > 3. Another thing that disappoints me is that the following timing result seems not reasonable: "the fastest setting of LKH (with only one search trial) takes 330 seconds for 10000 TSP100 instances, while AM and GANCO-AM (greedy) takes about 2 seconds on our machine." Solving 10000 TSP100 instances by AM means that you need to take $100\times 10000 = 10^6$ actions in total which means $10^6$ forward passes are finished in 2 seconds, i.e. each forward pass takes $2\times 10^{-6}$ seconds. I feel it is unreasonable that whether the forward pass will be that fast, and I take a toy experiment by counting the time of forward pass of a $10\times10$ linear layer on my 2080Ti, and it takes $3.8\times 10^{-5}$ seconds for a single forward pass. The forward pass of AM can only be much more complicated compared to my toy experiment, thus the result reported by the authors is not reliable. **If the authors can not clarify this potential misbehavior, I am also tending to further decrease my score.**
> >
> > 4. Concerning the LKH configuration, since LKH does not take so much time if it is configured properly, why not update with a new LKH entry and report the time and objective scores?
> >
> > 5. Though the time is cut compared to the original training time of AM, 3 days of generalization training still seems kind of too time-consuming.
> >
> > In summary, the authors do not fully address my concerns about this paper because this paper is studying the overfitting issue for underfitting models. Besides, the potential misbehavior about the running time also disappointed me. It even starts making me feel questionable whether the reported experiment results in the paper can be reproduced. I am not convinced that this paper should be accepted with the current version, and the authors should take it seriously about potential academic misbehaviors.

---

> > > ### Author Response · Authors · 2021-11-23
> > > **Response (2/2)**
> > >
> > > 5. Training deep learning models for COPs usually takes several days. And our goal is to offline train the model for fast online testing. In particular, our proposed framework is to generate one model for various distributions, which can save training time compared to training different models for different distributions separately.
> > >  Furthermore, as shown in Figure 2, similar to other deep learning methods, the performance improves fast at the beginning of training, where we can trade-off some performance for fast training with less adversarial iterations.
> > >
> > > [Kool2019] Wouter Kool, Herke van Hoof, and Max Welling. Attention, learn to solve routing problems! In International Conference on Learning Representations, 2019.
> > >
> > > [Kwon2020] Yeong-Dae Kwon, Jinho Choo, Byoungjip Kim, Iljoo Yoon, Youngjune Gwon, and Seungjai Min. Pomo: Policy optimization with multiple optima for reinforcement learning. Advances in Neural Information Processing Systems, 33, 2020.
> > >
> > > [Helsguan2017] Keld Helsgaun. An extension of the lin-kernighan-helsgaun tsp solver for constrained traveling salesman and vehicle routing problems. Roskilde: Roskilde University, 2017.

---

> > > ### Author Response · Authors · 2021-11-23
> > > **Response (1/2)**
> > >
> > > Thank you for your comment. We are glad that some of your concerns have been addressed. We hope that the following response would address the rest.
> > >
> > > 1. We totally agree that the performance drop would mean losing revenue for the company. However, please consider our target application, where the testing instances follow various unknown distributions. Therefore, the performance directly related to the revenue is the average performance on different distributions instead of one single distribution. We use the relative difference of optimality gaps to show the improvement of the GANCO framework. However, in terms of average objective value which is more directly related to revenue, GANCO-AM increases by 0.009 on the base training distribution and decreases by 0.066-1.750 (0.449 on average) on the five generalization distributions for TSP100. This means that GANCO-AM achieves better objective values even if the testing instances follow the base distribution with 95\% probability and the generalization distributions with only 5\% probability.
> > >  Consistently improving all instances is usually impossible for the challenging field of COPs. Strong traditional algorithms such as LKH3 can not find best known or better solutions for all the instances, as reported in [Helsguan2017].
> > > 2. As we discussed above, the improvement of GANCO applied to current deep learning models like AM is significant. The performance would rather fluctuate on the base distribution instead of consistently drops for different problem settings (e.g., GANCO-AM improves the performance on the base distribution for TSP20, CVRP100, OP20, PCTSP20, PCTSP50, PCTSP100 with greedy decoding).
> > >  The relative fluctuation on the optimality gaps (5.56\% larger for TSP100) seems a little large because the deep learning models perform very well on the base distribution (i.e., the optimality gaps are already very small for base training distribution). The absolute performance drop on the optimality gap is small (0.12\%) for the base training distribution, especially compared to the improvement for the generalization distributions (1.14\%-54.08\%).
> > >  The performance of deep learning models on the training distribution is good enough for the less studied problems, which is superior or comparable to strong traditional algorithms. However, the generalization performance is poor, which needs additional techniques to improve it for applications discussed above.
> > >  We would like to note that the proposed GANCO framework improves the generalization ability of existing models instead of completely solving the issue of overfitting.
> > > 3. One important fact that might be overlooked in the toy experiments is that the instances are usually run in batches for deep learning models. As the proposed GANCO framework is model agnostic, the hyper-parameters and evaluation process for each model are the same as those in the original work unless stated otherwise. And you can find similar results in the original papers. For 10000 TSP100 instances with greedy decoding, AM takes 6s on one 1080Ti GPU reported in the original AM paper [Kool2019] and takes 2s (possibly on a single Titan RTX GPU) reported in POMO paper [Kwon2020].
> > >  AM is a well-studied model and the results have been reported in many works, which all show runtime in similar scale. As mentioned in your previous comment, GANCO-AM and AM have the same runtime due to the same model architecture (with the same number of construction steps). The code for AM is publically available and you will get similar runtime with the pretrained models.
> > >  Regarding the comparison between deep learning models on GPU and traditional algorithms on CPU, we understand that it is not fair to directly compare the runtime on different devices, as also acknowledged in existing works of neural combinatorial optimization. In this paper, we follow the convention of existing works such as [Kool2019, Kwon2020] that both deep learning models and traditional algorithms solve multiple instances in parallel for a testing dataset. As stated in our paper, we run 20 instances in parallel on a 28-core CPU for the traditional algorithms. The AM and GANCO-AM are trained and tested on a RTX-2080Ti GPU.
> > >  Our code and trained models will be released upon acceptance for reproducibility.
> > > 4. Similar to other existing papers, our goal is not to outperform strong traditional algorithms but to push the performance of deep learning models.
> > >  The results reported in our paper take a considerably large space and are more comprehensive than most of the related works. We tried to avoid overwhelming results with more traditional baselines. However, we will tune the configurations of each traditional algorithm to include a version with runtime close to the deep learning models for the comparison in Appendix.

---

> ### Author Response · Authors · 2021-11-17
> **Response (1/2)**
>
> We would like to thank you for your time and the valuable comments. We hope our response could clarify the concerns:
>
> For weakness,
>
> 1. We need to emphasize that different from most existing works, we are targeting a more challenging scenario where test instances may not follow the training distribution. Moreover, the test instances may be even unknown. (We need to clarify that we do not assume that the test distributions, e.g. clustered, uniform or diagonal, are known beforehand.) This is reasonable and practical because obviously, we cannot expect to know the actual instance distribution before testing. For example, we cannot guarantee that the model trained on business data from the last month will work well in the next month, if some changes happen in the environment. We could fine-tune the network for a single test instance, but this could take a large amount of time, which contradicts the basic motivation of deep learning based methods, i.e. offline learning for fast online solving. Our aim is to make full use of the offline training time to improve the performance on unexpected out-of-distribution test instances. As mentioned by other reviewers, such generalization performance is important for practical neural combinatorial optimizers.
>  The performance drop on the base distribution is very small compared to the improvement on the generalization distributions. Taking TSP100 as an example, compared to the original AM, GANCO-AM improves the performance by 21.82\% to 87.98\% on the generalization distributions, with only a 5.56\% drop on the base distribution. As the GANCO framework does not change the optimization model architecture or the total number of parameters, the slight drop of performance on the base training distribution is fairly acceptable considering the large improvement on the generalization distributions. In addition, the performance does not necessarily drop on the base distribution as shown by the results of GANCO-AM on TSP20 and CVRP100.
>
> 2. First, we need to emphasize that we did not make such an assumption on the model capacity. Our aim is to propose a general model agnostic framework, so as to improve the generalization performance of a given Neural Combinatorial Optimization (NCO) model. Results show that our method can indeed improve the generalization of two state-of-the-art NCO models, i.e., AM and POMO. The model capacity is out of the scope, and should not be a major focus of this paper. Second, while the results of NCO models in our experiments are inferior to traditional solvers, they only take a fraction of time in solving with reasonably small optimality gaps. Such quality-time trade-off is usual in hard combinatorial optimization problems.
>
> 3. Yes, there is a special design. As described in the 5th paragraph (Optimization Training) in Section 3.2, we use a relatively simple data composition heuristic to keep some training samples from the base distribution and the previously generated distributions. This works well in our experiments. And our method is not particularly sensitive to the ratio of samples from the base distribution in the data composition heuristic, as the distributions dropped by much could be found by the following generation process.
>
> 4. We use 10000 trials as recommended in the AM paper [Kool2019] and adopted by many other related works. As shown in a very recent work [Xin2021], the fastest setting of LKH (with only one search trial) takes 330 seconds for 10000 TSP100 instances, while AM and GANCO-AM (greedy) takes about 2 seconds on our machine. Strong traditional heuristics like LKH start with almost completely random solutions and search a very large solution space based on their highly optimized searching scheme, therefore usually require a relatively longer time.
>  As we acknowledged in the 3rd paragraph of Section 4.5, current deep NCO models still are relatively inferior to strong traditional algorithms on well-studied COPs in terms of optimality. Nevertheless, for less studied problems, deep NCO models could perform favorably against the traditional ones with better solutions and shorter runtime at the same time (e.g., inferring 10 trajectories on PCTSP50, GANCO-AM gives 1.31% average gap on the 10000 test instances with average run time of 8s, while ILS fast needs to run for 106s with 1.67% average gap. Please refer to Table 8 for details). Note that many practical COPs are less studied than standard problems such as TSP, since they could have complicated constraints and objective functions customized for the specific industrial application, which lack the highly optimized traditional algorithms.

---

### Official Review · Reviewer_N945 · 2021-11-01

**Correctness:** 4
**Technical Novelty And Significance:** 3
**Empirical Novelty And Significance:** 3
**Recommendation:** 6
**Confidence:** 4

**Main Review:**

Strengths:

- paper is well written
- good evaluation
- intuitively understandable method with good results, both in terms of generalisation and speeding up training (which is expected due to other works establishing selecting tough training instances helping convergence)

Weaknesses
- the concept reminds me strongly of https://ai.googleblog.com/2021/03/paired-new-multi-agent-approach-for.html which should be cited
- I wonder how much of the improvement is due to simply training on more examples? did you try generating random problem instances of multiple distributions and training AM for the same number of episodes? I'd still expect the adversarial method to be more universal, but if the distributions we might expect are somewhat known, domain randomisation can be computationally cheaper and simpler than this setup. Especially if a small subset of distributions can capture most of the expected variation across test distributions.

Other comments:

- instead of training in stages, did you attempt alternating the optimizer and the adversary on a per step basis?

**Summary Of The Paper:**

The paper proposes a way to generate training examples for neural combinatorial optimisation  (NCO) methods (specifically evaluated on POMO and AM) that leverages an adversarial RL policy trained in stages to generate instances that maximise the performance advantage of a baseline CO method over the NCO method. The method is tested on a variety of CO problems and shows improved OOD generalization especially for larger problem instances.

**Summary Of The Review:**

The paper presents an intuitive approach to generating training examples, makes it work and does a good evaluation. However, similar work has been done in the RL domain and other supervised training domains before (e.g. using GANs for data augmentations) so giving a marginal accept for now.  If the missing citation is added, my questions are answered and the edge the *adversarial* augmentation has over *random* augmentations (or simply perturbing the network slightly and training for longer) I'm willing to update my score.

---

> ### Author Response · Authors · 2021-11-17
> **Response**
>
> We would like to thank you for your time and the valuable comments. We hope our response could clarify the concerns.
>
> For weakness,
>
> 1. Thank you for pointing this out. We have added the citation [Dennis2020] and discussion in the related works section of the revised version.
>
> 2. In our experiments, the base model is trained on the base distribution until convergence. Further training will not improve the performance on the base distribution or the generalization distribution, as shown in Appendix H.
>  Thank you for your suggestion on domain randomisation. We have conducted a baseline experiment that trains AM (after the pretraining stage) on multiple distributions. We demonstrate that GANCO-AM achieves most of the benefits of training on in-distribution data and achieves better generalization performances compared to the model trained with multiple distributions. Please refer to Appendix I for details.
>
> For other comments:
>
> 1. At the beginning of each generation stage, the parameters of the generation model are re-initialized randomly. In doing so, it prevents the model from being stuck in the local optimum of previous adversarial iterations and also allows the model to learn diverse instance distributions. And the optimization models are trained on the hard distributions generated by the generation model at the end of each generation training.
>  Alternating the optimizer and the adversary on a per-step basis would mean that the optimization model will also be trained on some relatively easy distributions which the generation model finds during the exploration phase. And we will need some techniques to help the generation model escape local optimum (such as periodically re-initializing the generation model parameters). In our experience, this way could work but probably will converge slower due to the unnecessary training on some easy distributions.
>
> [Dennis2020] Dennis M, Jaques N, Vinitsky E, et al. Emergent Complexity and Zero-shot Transfer via Unsupervised Environment Design[J]. Advances in Neural Information Processing Systems, 2020, 33: 13049-13061. MLA  (https://ai.googleblog.com/2021/03/paired-new-multi-agent-approach-for.html)

---

> > ### Comment · Reviewer_N945 · 2021-11-21
> > **Thank your for the additions**
> >
> > Thank you for the additions, this has alleviated all of my concerns, barring any concerns raised in the discussion with other reviewers I'm open to increase my score to a 7 now.

---

> > > ### Author Response · Authors · 2021-11-21
> > > **Response**
> > >
> > > Thank you very much for acknowledging the value of our paper with a quick response.

---

### Official Review · Reviewer_tjCH · 2021-11-02

**Correctness:** 2
**Technical Novelty And Significance:** 2
**Empirical Novelty And Significance:** 2
**Recommendation:** 3
**Confidence:** 3

**Main Review:**

The paper is reasonably well written, though section 3.1 would be easier to follow if the architecture of the generator model was described using a figure instead of just text. Moreover, since the authors evaluate their approach on optimizer models trained through reinforcement learning (according to section 2), I would have liked to see a discussion of other approaches that have been taken to improve the robustness of RL techniques in the related work section.

The evaluation section demonstrates empirically that data augmentation helps improve the performance of ML based combinatorial optimization techniques: it covers several enough combinatorial optimization problems (e.g TSP, CVRP, OP, PCTSP, KP) to give confidence to the reader that it is very likely to succeed on any combinatorial optimization problem. Furthermore, it evaluates the effectiveness of data augmentation in conjunction with 2 common optimizer models, AM and POMO, which indicates that data augmentation can be leveraged to improve the effectiveness of several prior work in this area.

However, it is widely known that data augmentation helps the performance of neural networks, and the authors fail to demonstrate the effectiveness of their particular flavor of data augmentation. At the very least, they should have compared their approach against random data augmentation as well as against a simple policy that generates n examples, and keeps the k% of these n examples on which the optimizer model performs the worst. In other words, the paper needs to demonstrate that the generator model truly learns how to improve the quality of the dataset on which the optimizer model is trained. Put another way, the authors explain very well what they set out to accomplish in figure 2. They need to demonstrate in their evaluation that they accomplished this.


**Summary Of The Paper:**

Over the last few years, several machine learning based approaches have been proposed to solve common combinatorial optimization problems, such as the traveling salesman problem. However, it's never been entirely clear how well the previous work generalized to out of distribution instances. This paper proposes to improve generalization through dataset augmentation.
Specifically, the paper proposes to train a model (aka the generator) that is used to drive the creation of training instances on which the neural network used to solve the combinatorial problem (aka the optimizer) does poorly. The authors then show that training the optimizer model on a set of examples taken from the original dataset as well as created under the guidance of the generator model results in better performance than training the optimizer model on the original dataset only.

**Summary Of The Review:**

This paper attempted to tackle an important problem which hasn't received the attention it deserved, namely ensuring that neural networks trained to solve combinatorial optimization problems generalize well. They proposed a theoretically sound approach to augment the original dataset with additional training examples created under the guidance of an auxiliary 'generator' model to cover the gaps in the original dataset. However, they failed to demonstrate that the effectiveness of this 'generator' based approach.

---

> ### Author Response · Authors · 2021-11-17
> **Response**
>
> We would like to thank you for your time and the valuable comments. We hope our response could clarify the concerns:
>
> We have added a discussion of other approaches that have been taken to improve the robustness of RL techniques in the related work section. And we will revise Figure 1 to include the architecture of the generator model.
>
> We agree that a simple policy (that generates n examples, and keeps the k% of these n examples on which the optimizer model performs the worst) is a very good and convincing baseline. In fact, we have compared with a baseline following a very similar idea in our experiments. We use the genetic based crossover and mutation operators following [Liu2020] to randomly generate new instances and keep the hard instances which the optimizer model performs the worst. Please refer to the 4th and 5th paragraph in Appendix A for details and the result comparison, which show that our method significantly outperforms this strategy. The reason is that we design a generation model that explicitly discovers the hard distribution beyond the base distribution, trained by the adversarial process. In contrast, the simple strategy of randomly generating instances and keeping the hardest ones could be much less effective in finding the actual hard instances, because it lacks the ability of generating instances outside the base distribution.
>
> In the revised version, we have also followed the suggestions from other reviewers to conduct more experiments to train the baseline model on instances sampled from multiple distributions. We demonstrate that GANCO-AM achieves most of the benefits of training on in-distribution data and achieves better generalization performances compared to the model trained with multiple distributions. Please refer to Appendix I for details.
>
> [Liu2020] Shengcai Liu, Ke Tang, and Xin Yao. Generative adversarial construction of parallel portfolios. IEEE transactions on cybernetics, 2020.

---

### Official Review · Reviewer_sEuD · 2021-11-02

**Correctness:** 3
**Technical Novelty And Significance:** 3
**Empirical Novelty And Significance:** 3
**Recommendation:** 6
**Confidence:** 4

**Main Review:**

**Strengths**

- The paper is well written, the framework is presented clearly and (along with the supporting code promised to be released) with sufficient information to reproduce the key elements if the work.

-  I find the task itself, improving generalisation in RL COP solvers, to be well motivated and prescient.  Indeed, most works learning solvers for combinatorial problems seek to test how they perform when the test-time distribution is different from that on which the solver was trained and I find the idea of dynamically controlling the training distribution to challenge the learnt heuristic to be an interesting approach.  To the best of my knowledge, the proposed framework is novel.

- The experimental results are impressive, with the improved generalisation of agents trained within the GANCO framework clearly demonstrated with respect to those trained on only a single base distribution.  By considering multiple COP instances and using established methodologies, the empirical performance improvements are clear, however it there remains scope to further improve the analysis of GANCO itself as discussed below.

**Weaknesses**

- Whilst the broader distribution of training tasks provided by GANCO is clearly demonstrated to improve generalisation, the necessity of the adversarial entity (GANCO’s core contribution) is not demonstrated clearly enough to unequivocally endorse GANCO as an optimal (or very good) approach to generating additional instances.  To be precise, the RL baselines to which GANCO is compared are trained on strictly fewer instances sampled from a narrower distribution and for fewer steps overall, therefore it is unsurprising that GANCO performs better across different distributions at test time.  Whilst the authors do compare GANCO to a genetic based (GB) method for generating data, they also highlight that it is unsuitable for the task at hand: "The resulting distribution [from GB] still approximately follows the base training distribution. It would be extremely hard for the genetic operators to attain instances following a significantly different distribution.”.  Given GANCO requires training a second adversarial agent (which comes with a significant overhead),  I believe justifying this additional training as necessary is important.  At the least, I would like to see comparisons to:

 	1. The performance on an RL agent trained on multiple/all target distributions.  GANCO would, of course, not be expected to beat this, however it is unclear from the current results if GANCO is achieving most of the benefits of training on in-distribution data, or only providing fractional improvement compared to what could be achieved.

	2. The performance of an agent trained on the curriculum of instances generated by GANCO *from a different training run*.  It would be highly insightful to know whether it is the ability of GANCO to tailor instances bespoke to the current agent, or it is simply the ability to generate many significantly different distributions from the base distribution, that provides the bulk of the performance improvement.  If it was demonstrated that the adaptive nature of GANCO is key, this would, in my opinion, significantly strengthen the paper and further justify the design choices of the framework.

- I do not believe Figure 2 is suitable as it is a set of ‘plots’ denoting how the authors hope GANCO will change the performance across a distribution (e.g. performance is strictly improved for all instances and all stages of the pipeline), and does not present any data or additional insight.  The intuition that providing additional data from distributions on which the agent performed poorly and help to learn more general policies is clear and suitably presented in the rest of the paper without the need for Figure 2, therefore I would suggest removing it.

**Additional comments, question and errata**

- The authors state that both the inputs to the generating network are sampled noise, and that the outputs are also per-node distributions from which node attributes are sampled.  What is the purpose of have two sources of stochasticity?  Initially I interpreted the input noise as sampling the specific problem instances distribution, and the output as sampling a value from this distribution.  However as the attributes for each node are sampled sequentially, presumably the input noise is resampled at each state too?  Of course it is not a major issue, but I would appreciate clarity for my own insight!

- The authors claim that the challenge of generalising to out-of-distribution (OOD) test-time instances is “particularly severe for neural combinatorial optimization (NCO) models because the solution quality intricately depends on the instance distributions”.  Whilst I would agree that OOD generalisation is challenging and important, it is not clear to be that NCO models find it especially difficult compared to the multitude of other ML/RL applications.  Indeed, as solving COPs is often extremely (NP-)hard, an alternative stance would be that ML4CO models will never learn to analytically solve even in-distribution tasks of meaningful size, an so will instead naturally rely on sub-optimal heuristics that may in fact generalise quite well (e.g. a greedy algorithm is simple to learn, sub-optimal, but often shows good generalisation).  Could the authors clarify their meaning, or alternatively simply acknowledge that generalisation is difficult and important in general, with COPs being no exception?

- Is there a particular reason the authors settled on alternately training the solver and generator, as a naive implementation might train both jointly (or indeed train exclusively on the generated instances)?

- Typo in the caption of Fig 3c: (c) “CANCO-POMO on CVRP100”.


**Summary Of The Paper:**

This work presents GANCO, a framework that extends the training of RL heuristics for combinatorial optimisation problems (COPs) to include an adversarial agent that aims to generate hard-to-solve problem instances.  The RL heuristic and adversarial instance generation are alternately trained, with the intuition that the adaptive problem distribution encourages the solving agent to learn policies with better generalisation performance to unseen instances compared to those trained on a single static distribution.  Experiments train the Attention Model of Kool et al (and a subsequent extension, POMO) within the GANCO framework and show that across multiple well-studied COPs (including the travelling salesman problem), GANCO augmented training does result in policies that generalise better to unseen distributions at test time.

**Summary Of The Review:**

Whilst the GANCO framework is novel as a whole, it is the combination of existing ideas and methodologies (e.g, learning construction heuristics for CO problem and adversarial training).  The field of ML4CO is active the particular problem considered in the is work, generalising to OOD test-time data, is ubiquitous - therefore, in my opinion, the paper comfortably reaches the required standard in terms of novelty and potential interest.  Ultimately then, I believe it comes down to whether the demonstrated empirical performance is sufficiently strong, or if the core idea of adversarial instance generation is sufficiently well justified and investigated.  My feeling is that it is not clear cut, with the experimental results impressive, but to me only unequivocally demonstrating that training on a broader distribution of tasks results in a more general agent.  Overall, however, there is enough justification of the GANCO framework in the latter sections, for me to lean towards acceptance (ultimately, I find the ideas and results in the paper interesting, and therefore feel the same can be said for others working in the field).  With that said, I would be more confident and comfortable in my recommendation if the adaptive nature of the adversarial instance generation that is core to GANCO could be demonstrated as a key element in the frameworks empirical success (as discussed in detail in my main review).

---

> ### Author Response · Authors · 2021-11-17
> **Response (2/2)**
>
> 1. (2) (Continued) To demonstrate the adaptive nature of GANCO’s adversarial instance generation, we conduct the experiments of training the agent for one problem (OP) on the distribution curriculum generated by GANCO for another agent for a different problem (PCTSP), namely, GANCO-AM-DIFF. The average performance of GANCO-AM-DIFF on the five generalization distributions is 43.26, 43.55 and 43.90 with 5, 10 and 20 adversarial iterations (the larger the better for OP). In comparison, results of the original GANCO-AM for OP are 43.41, 43.73 and 44.00, respectively. Result of the original AM is 41.02. We can see that the performance of GANCO-AM-DIFF is also pretty good as these two problems are very similar with the distinction of flipping the constraint and objective value. However, the original GANCO-AM for OP performs better, suggesting that the generation model indeed tailors distributions bespoke to the specific problem. The experiment has been added in Appendix I of the revised version.
>
> 2. Thank you for pointing this out. We tried to use this plot as an informal illustration to demonstrate the ideal process following Figure 1 in [Goodfellow2014]. However, the assumption (that the performance is strictly improved for all instances) certainly is not realistic. We have removed this plot and used the saved place to add our new baseline experiments in the revised version.
>
> For additional comments, question and errata,
>
> 1. Your initial interpretation is actually very accurate (the input noise as sampling the specific problem instance distributions, and the output as sampling a value from this distribution). The generation model will output the distribution parameters for different nodes at the same time, based on which we sample the node attributes. Therefore, the input noise is the same at each state and we can also view the action as sampling all node attributes at the same time. The description of action as generating attributes for one node sequentially is misleading in hindsight and we have rephrased it in the revised version to be accurate.
>  Actually, we specifically use two sources of stochasticity and both are necessary. It would be easier to understand if each round of generation process is viewed as generating instance distributions instead of generating instances from a distribution. We use the noise input to generate different distributions (each noise input will correspond to a different distribution). Therefore, each round of generation task will generate a large number of distributions for the optimization model. And due to the reinforcement learning training, we need to sample from the output distribution to explore.
>
> 2. By a particularly severe challenge for NCO models, we mean that deep learning models for COPS are mostly trained on instances sampled from specific distributions. And the solution patterns could be very different for instances from different distributions. As shown in our experiments, the generalization performance could be quite poor for NCO models.
>  However, this statement also applies to the RL trained models for the game environments. And we do not have a specific measure for the generalization ability on different tasks as this highly depends on the training and testing datasets. We have rephrased this part to be more accurate in the revised version.
>
> 3. At the beginning of each adversarial iteration, the parameters of the generation model are re-initialized randomly. In doing so, it prevents the model from being stuck in the local optimum of previous adversarial iterations and also allows the model to learn diverse instance distributions. And the optimization models are trained on the hard distributions generated by the generation model at the end of each generation training.
>  Joint training would mean that the optimization model will also be trained on some relatively easy distributions which the generation model finds during the exploration phase. In our experience, joint training could work but probably will converge slower due to the unnecessary training on some easy distributions. And we will need some techniques to help the generation model escape local optimum.
>  Instead of training exclusively on the generated instances, we train the optimization model with a data composition rule to achieve desirable performance on both the base training distributions and the adversarial generated distributions. Please refer to the 5th paragraph (Optimization Training) in Section 3.2 for details.
>
> 4. Thank you for pointing out the typo and we have fixed it.
>
> [Liu2020] Shengcai Liu,  Ke Tang,  and Xin Yao.   Generative adversarial construction of parallel portfolios.IEEE transactions on cybernetics, 2020.
>
> [Goodfellow2014] Ian Goodfellow, Jean Pouget-Abadie, Mehdi Mirza, Bing Xu, David Warde-Farley, Sherjil Ozair, Aaron Courville, and Yoshua Bengio. Generative adversarial nets. Advances in neural information processing systems, 27, 2014.

---

> ### Author Response · Authors · 2021-11-17
> **Response (1/2)**
>
> We would like to thank you for your time and the valuable comments. We hope our response could clarify the concerns:
>
> For weakness,
>
> 1. Regarding “strictly fewer instances and fewer steps”, in Table 11 from Appendix H, we have shown that training AM with more instances and steps until convergence does not help in improving its generalization ability. The generalization performance fluctuates with more training epochs instead of improves. In the revised version, to be more fair, we further compare GANCO-AM against a variant of AM named AM-S, which is trained on the base distribution for the same number of steps (also the same number of instances) after the pretraining stage as GANCO-AM. Taking TSP50 as an example, the average performance of AM (after the pretraining stage), AM-S and GANCO-AM on the five generalization distributions is 4.156, 4.168 and 3.989, respectively (the smaller the better). The generalization ability does not improve with more epochs for the original AM. And the improvement of GANCO over the other two models is clearly significant. Please refer to Appendix H and Table 11 in the revised version for more details.
>  Regarding our comment on GB, we mean that this traditional method in [Liu2020] to generate hard instances fails to solve the target challenge of generalizing deep learning models to different distributions. With the best of our effort, we did not find more suitable methods in the literature which generate data samples to improve the generalization ability of deep NCO models.
> Regarding the two required experiments,
>  (1) Following your suggestion, we conduct experiments of training the original AM on 4 distributions (base, uniform, diagonal, gaussian) after the pre-training stage, namely, AM-T4. The training dataset equally samples from the 4 distributions and AM-T4 is trained for the same number of steps (also the same number of instances) as the GANCO adversarial training stage. Please refer to Appendix I for detailed results. For TSP100, the average performance of the original AM (after the pretraining stage), AM-T4 and GANCO-AM is 6.26, 5.82 and 5.75 on the four AM-T4 training distributions and 5.83, 5.79 and 5.73 on the other two generalization distributions (the smaller the better). For OP100, the average performance of the original AM, AM-T4 and GANCO-AM is 38.47, 41.22 and 41.07 on the four AM-T4 training distributions and 41.82, 42.90 and 44.02 on the other two generalization distributions (larger collected prizes are better).
>  On the four distributions AM-T4 is trained on, GANCO-AM achieves comparable or better performance than AM-T4. The performance of GANCO-AM is better than AM-T4 on some distributions even though AM-T4 trains on these distributions. And AM-T4 underperforms the original AM on one AM-T4 training distribution (gaussian) for TSP100. This is because equally sampling from four distributions leads to unbalanced reward (the rewards for instances with some distributions dominate the gradients). Instead of equally sampling from each of the four distributions, we also tried another simple composition rule (1/2 base distribution and 1/6 for each of the three generalization distributions) and the results are similar. In contrast, our GANCO framework uses the alternative training process to dynamically find the hard distributions and adjust the proportion of newly generated distributions in the training dataset during the adversarial training stage. Compared to AM-T4, GANCO-AM achieves most of the benefits of training on in-distribution data.
>  More importantly, on the two generalization distributions which both models do not train on, the average performance of GANCO-AM is much better, which also demonstrates the improvement of generalization ability by training on adversarially generated distributions.
>  (2) In our experiments, pre-training the same agents with different runs results in similar agents, which perform similarly on the same distribution (i.e., the hard distributions for one agent are usually hard for the other agents). Therefore, the performance of an agent trained on the curriculum of instances generated by GANCO from a different training run is also pretty good. However, this is due to the inherent similarity of the pretrained agents, and does not mean the improvement of GANCO is simply because the model is trained on more different distributions.

---

> ### Comment · Reviewer_sEuD · 2021-11-22
> **Response to rebuttal (pre-discussion)**
>
> Thank you for the detailed response to my questions and comments, as well as the notable effort to perform additional experiments.
>
> The only point on which I am not satisfied is the results of training GANCO on a curriculum from a different run.  It seems like there is very little, if any, improvement attributable to GANCO tailoring the training distribution to specific agent being trained.  Instead, the results suggest to me that GANCO is essentially learning a general data augmentation, which can then be re-used across agent initialisations.  Whilst the authors justify this by stating that pre-training results in sufficiently similar agents that bespoke tailoring is not required, this seems to undermine the need for the extensive GANCO framework.  Indeed, the difference in performance when using a curriculum learnt for a different COP is minor and, to me, not significant enough evidence of the adaptive nature of adversarial training.
>
> I will wait for the discussion period before committing to a score, however with this concern (and those raised by other reviewers) I am not comfortable increasing my score at this point.  Indeed, I would consider reducing it to a 5 if the GANCO framework can not be shown to benefit from the agent-specific adaption of adversarial training.

---

> > ### Author Response · Authors · 2021-11-25
> > **Response**
> >
> > Thank you for your response. The difference in performance when using a curriculum learnt for a different COP is relatively minor because the two problems are very similar with the only distinction of flipping constraints and objective values. We agree that tailoring to different agents is a desirable property, but it is not the only measure for the effectiveness of the proposed framework.
> >  As shown in the experiments of AM trained with multiple distributions, the generalization performance is much worse than GANCO-AM, suggesting that simple data augmentation is not sufficient. We do lack an effective method to generate instance distributions for training models with good generalization performance for COPs. GANCO framework can find effective curriculum of distributions. Even though the distributions generated for one problem also work well for another problem (due to inherent similarities between the problems), the distributions and the number of instances for each distribution in the training dataset are not straightforward to determine.

---

### Comment · Reviewer_mumN · 2021-12-02
**The time measurement may be unfair in (Kool2019, Kwon2020) and this paper**

Dear all,

Recently I come up with the concern about whether the measurement of time is proper in a line of MLCO papers (Kool2019, Kwon2020) including this paper. In these papers, the MLCO algorithms are run in parallel on GPU, and the time reported is the time it takes to solve a single problem if batch=1, divided by the batch size. However, their compared baselines that run on CPU, are not run in parallel batches. I believe this is not a fair comparison because the number of parallel batches is different, and in practice, it is easy to solve multiple problems in parallel batches with multiple CPU nodes. In contrast, what we should care about is the time it takes to solve a single problem with batch=1 because it corresponds to how long we will keep the user waiting for the optimization result.

Though the discussion period is scheduled to have been ended, I am writing this comment to raise the concern because this seems to be an issue even in some highly-cited papers. The authors of this paper may have simply followed the open-source code from others and did not think carefully about the timing results. However, it does not mean that the authors should not be criticized for conducting an unfair comparison among different methods, and it is the responsibility of we researchers to stop delivering misleading information to our future readers.

[Kool2019] Wouter Kool, Herke van Hoof, and Max Welling. Attention, learn to solve routing problems! In International Conference on Learning Representations, 2019.

[Kwon2020] Yeong-Dae Kwon, Jinho Choo, Byoungjip Kim, Iljoo Yoon, Youngjune Gwon, and Seungjai Min. Pomo: Policy optimization with multiple optima for reinforcement learning. Advances in Neural Information Processing Systems, 33, 2020.

---

> ### Comment · Reviewer_N945 · 2021-12-02
> **I agree with the spirit of the comment but would suggest a different type of fairness**
>
> While I agree with the concern raised that comparing a fully GPU batch vs batchsize 1 on CPU, I think the proper comparison would be to observe the scaling behaviour on some representative compute node (say amazon P3 and C5 instances) with the same dollar budget and then going across various batch size settings. Alternatively, one could keep track of the total numbers of FLOPS per problem. Constraining GPUs  running Deep Algorithms to run with batchsize 1 is about as unfair as constraining traditional solvers to run with batchsize 1 on a single node, as
>
> - the python machine learning framework and python loop vs. the (often highly optimised) C/C++ code make direct runtime comparison difficult
> - GPUs are always going to be slower than CPUs for non-parallel workloads due to the type of parallelisation, so as long as the termination time is "reasonable"^TM, one should try to account for the ability to solve things in parallel (one can easily imagine the value of solving many different problems and variations of problems in one go).
>
> Comparing the "problems per $" or "FLOPS/problem" metrics and reporting scaling results would (I think) fairly capture your concerns as well as the inherent tradeoffs involved in choosing these methods.

---

### Decision · Program_Chairs · 2022-01-20

**Decision:**

Reject

**Comment:**

## A Brief Summary
Recent works in deep learning have shown that it is possible to solve [[combinatorial optimization]] problems (COP) with neural networks.  However, generalization beyond the examples seen in the training set is still challenging, e.g., generalizing to TSP with more cities than the ones seen in the training set. This paper proposes the GANCO approach, where a separate generative neural network based on GAN generates new hard-to-solve training instances for the optimizer. The optimizer and the generative network are trained in an alternating fashion. The authors have run experiments with the attention model (AM) and POMO with their GAN-based data augmentation approach. The authors provide experimental results on several well-known COPs, such as the traveling salesman problem.

## Reviewers' Feedback

Below, I will summarize some reviewers' feedback and would like the authors to address the cons noted below.
### Reviewer sEuD

*Pros:*
- Paper is well-written.
- Task is important and well-motivated.
- Good experimental results.
*Cons:*
- The paper's core contribution on the necessity of adversarial entities is not well-motivated.
- Missing baselines:
- RL agent trained on all target distributions. To figure out how far GANCO is from the optimal policy.
- The performance of an agent trained on a curriculum.
- Figure 2 is unnecessary/redundant in the paper.



### Reviewer tjCH
*Cons:*
- The paper is reasonably written. However, it would be much easier to follow with a few changes. For example, section 3.1 explains the architecture and, in related works, a more comprehensive overview of the methods to improve the robustness of RL methods.
- It is widely known that data augmentation helps in deep learning. The paper's claims would be more convincing if it provided some crucial baselines, such as comparing different data augmentations methods and carefully ablating them.

### Reviewer N945
*Pros:*
- Well-written
- Good evaluation
- Simple model with good results
*Cons:*
- Missing citation to the PAIRED paper.
- How important are the adversarial entities generated? Is it possible to achieve similar results by just training on more samples?
- Missing baselines: Instead of training in stages, alternate optimizer and generator network per step basis.

### Reviewer mumN
*Pros:*
- The proposed approach is novel.
- Comprehensive and extensive experiments.
- Figure 1 provides a good summary of the approach.

*Cons:*
- Motivation is for the GANCO is not very convincing.
- Concerns about the capacity of the neural nets used in the paper.
- Concerns on forgetting the original distribution.
- Concerns about experimental evaluation protocol.
- Including experiments on routing problems to show the generality of the proposed approach.
- Request for improvements in the writing and the formatting of the paper.

## Key Takeaways and Thoughts
I think this paper attacks an interesting problem. As far as I am aware of the approach is novel. However, generative adversarial networks have been used in the machine learning literature for data augmentation and RL for augmenting the environment (see the PAIRED paper.) GAN type of approaches hasn't been used to improve the generalization of the deep learning approaches for COP. The results look promising. However, as pointed out by Reviewer mumN and tjCH, this paper would benefit more from further ablations, particularly the necessity of adversarial generation part to make the arguments more convincing. As it stands now, it is not clear where exactly the improvements are coming from. Reviewer mumN also raised some concerns about the poorly configured LHK3 baseline in the discussion period. Furthermore, I agree with the reviewer mumN and tjcH that this paper would benefit from restructuring to make it flow better. I do think that this paper needs another round of reviews. I would recommend the authors go over the feedback provided here and address them for future submission.## References